# Retinoid Synthesis Regulation by Retinal Cells in Health and Disease

**DOI:** 10.3390/cells13100871

**Published:** 2024-05-18

**Authors:** Massimiliano Andreazzoli, Biancamaria Longoni, Debora Angeloni, Gian Carlo Demontis

**Affiliations:** 1Department of Biology, University of Pisa, 56126 Pisa, Italy; 2Department of Translational Medicine and New Technologies in Medicine, University of Pisa, 56126 Pisa, Italy; 3The Institute of Biorobotics, Scuola Superiore Sant’Anna, 56127 Pisa, Italy; 4Department of Pharmacy, University of Pisa, 56126 Pisa, Italy

**Keywords:** vision, rod photoreceptors, cone photoreceptors, RPE, Müller glial cells, 11cis retinaldehyde, 9cis retinaldehyde, retinoic acid, RPE65, retinal dystrophies

## Abstract

Vision starts in retinal photoreceptors when specialized proteins (opsins) sense photons via their covalently bonded vitamin A derivative 11cis retinaldehyde (11cis-RAL). The reaction of non-enzymatic aldehydes with amino groups lacks specificity, and the reaction products may trigger cell damage. However, the reduced synthesis of 11cis-RAL results in photoreceptor demise and suggests the need for careful control over 11cis-RAL handling by retinal cells. This perspective focuses on retinoid(s) synthesis, their control in the adult retina, and their role during retina development. It also explores the potential importance of 9cis vitamin A derivatives in regulating retinoid synthesis and their impact on photoreceptor development and survival. Additionally, recent advancements suggesting the pivotal nature of retinoid synthesis regulation for cone cell viability are discussed.

## 1. Introduction

Living organisms may sense environmental changes in light intensities using specialized proteins, the opsins. Complex eyes combining opsin-expressing photoreceptor cells with optical structures (lens and cornea) enable the transition from light-sensing to image-forming vision. Eye organization may differ significantly between vertebrates and invertebrates at anatomical and molecular levels (reviewed in [1]). The initial distinction between rhabdomeric and ciliary photoreceptors, as typical of invertebrates and vertebrates, respectively, has been blurred by new evidence showing they coexist in the marine annelids of the *Platynereis* genus [2], with rhabdomeric photoreceptors in the eye and ciliary photoreceptors in the brain. Furthermore, ciliary photoreceptors in *Platynereis* and humans express a similar opsin [2], suggesting rhabdomeric and ciliary photoreceptors evolved from a common ancestor (reviewed in [3]). Rhabdomeric and ciliary photoreceptors sense light using structurally diverse opsins indicated as rhabdomeric-(r-opsins)- or ciliary-type (c-opsins) (reviewed in [3]). 

Despite their structural differences, at the heart of light-sensing by both r- and c-opsins, there is a covalently bonded vitamin A derivative, 11cis retinaldehyde (11cis-RAL). Light triggers the isomerization of the double bond of the 11cis-RAL isoprenyl chain to all-trans RAL (at-RAL), leading to opsin conformational changes that activate the phototransduction cascades specific to r- or c-type photoreceptors (reviewed in [1]). Activation of the phototransduction cascade will depolarize r-type photoreceptors [4] or hyperpolarize c-type photoreceptors [5]. Opsins must revert to the 11cis-RAL-bound state to regain their ability to respond to light, but 11cis-RAL regeneration from at-RAL occurs via different mechanisms in r- and c-type opsins. For r-opsins, 11cis-RAL regeneration occurs by photoconversion, following the absorption of a second photon by at-RAL. As shown for melanopsin, an r-opsin expressed by intrinsically photosensitive retinal ganglion cells (ipRGC) [6,7] of the vertebrate retina, 11cis-RAL regeneration depends on an intrinsic photoisomerase activity triggered by the absorption of a photon of the longer wavelengths of those triggering its activation [8] (summarized in [9]). Therefore, similar to r-opsins, melanopsin regeneration does not require ex novo 11cis-RAL synthesis by a dedicated enzymatic pathway. The reduction in the ipRGC-mediated response in mice lacking the ability to generate ex novo 11cis-RAL may represent a secondary effect of faulty ciliary photoreceptors on ipRGC development [10].

At variance with r-opsin, c-opsins of ciliary photoreceptors do not revert to their ground state upon light stimulation. Consequently, c-photoreceptors require a specialized biochemical pathway for regenerating 11cis-RAL from at-RAL. This pathway, known as the retinoid cycle, operates in darkness and has been reviewed in [11]. The ability to regenerate 11cis-RAL in darkness through the retinoid cycle has several advantages that have been previously covered [3]. However, it may require careful control, as either reduced or increased 11cis-RAL synthesis is associated with adverse effects on photoreceptor viability [12].

Furthermore, after cells of the retina pigment epithelium (RPE) were found to express a 65 kDa protein (RPE65) that was identified as the isomerohydrolase introducing the 11cis double bond in the retinoid cycle in the retinal pigment epithelium (RPE) cells [13,14], novel evidence indicates 11cis-RAL synthesis via an additional enzyme in RPE cells and in Müller glial cells (MGCs) (reviewed in [12,15]). However, the evidence on the control of 11cis-RAL and its functional relevance has not been recently covered. In addition, recent evidence shows retinoic acid’s role in rod and cone receptor viability, but the possible role of 9cis derivatives of vitamin A has not been addressed so far. We aim to provide a new perspective on retinoids’ roles in retinal health and disease by discussing these points.

## 2. The Retinoid Cycle

Ciliary photoreceptors and rhabdomeric photoreceptors have similar response kinetics and sensitivity to light (discussed in [16]). In addition, r-opsins can quickly recover to their ground state by the photoconversion of at-RAL to 11cis-RAL, making one wonder whether ciliary photoreceptors have been the best choice for image-forming vision by vertebrates. Considering this hypothetical question, one should be aware that the photoconversion of at-RAL to 11cis-RAL has a critical drawback, i.e., wavelength-dependent light absorption by water [17], which restricts the vision of marine organisms in relatively shallow water. Specifically, water absorbs light in the yellow spectral region (565–590 nm) 10 times more effectively than in the blue (420–450 nm) spectral region [17], indicating that, below 50 m depth, blue light may lead to 11cis-RAL isomerization to at-RAL, but not enough yellow light could be available to regenerate 11cis-RAL from at-RAL, i.e., preventing vision by animals with r-opsins eyes in water deeper than 30–40 m. 

Ciliary photoreceptors using c-opsin have lost the ability to use photoconversion to regenerate 11cis-RAL from at-RAL. As shown schematically in Figure 1A,B, the difference between c- and r-opsins has been tracked to a second negative charge provided by a highly conserved glutamate residue located at position 113 (E113) in bovine rhodopsin. 

The E113 negative charge stabilizes the protonated Schiff’s base positive charge, generated by the covalent bond between an opsin-conserved lysine and 11cis-RAL. However, following the light-induced cis-trans isomerization, E113 became protonated by the proton released from E181 [18] (reviewed in [3]). Following Schiff’s base deprotonation, at-RAL-bound c-opsin corresponding to Metharhodopsin II (MII) undergoes several structural changes leading to GTP-binding protein activation [19]. These MII and E113 protonation structural changes prevent c-opsins from returning to the 11cis-RAL-bound ground state. 

In c-type opsins, the recovery to the ground state with 11cis-RAL bound requires at-RAL dissociation from opsin. The released at-RAL is converted back to 11cis-RAL by a biochemical pathway indicated as the retinoid cycle, whose steps occur in photoreceptors and RPE cells. The spatial interplay between the four main cell types involved in the retinoid cycle is indicated in Figure 2. RPE cells generate 11cis-RAL, which reaches photoreceptors by diffusion across the subretinal space (SRS) (straight double-arrow line). As shown in Figure 2, the SRS is limited by junctional complexes (JC) between RPE cells and those between photoreceptor inner segments (IS) and the apical processes of MGCs. 

An advantage of the synthesis of new 11cis-RAL via the retinoid cycle (reviewed in [11]) is that it may occur in darkness, thus enabling vision in dim light, i.e., at moonlight levels [20] or in deeper water, overcoming the limitations faced by organisms equipped with photoreceptors using r-opsins. Rod-type ciliary photoreceptors represent the most common photoreceptor type in most vertebrates, including humans, and are optimized for dim light vision. Rod photoreceptors have two additional advantages. First, disks in their outer segment (OS) prevent rhodopsin diffusion over the entire OS length, so the rhythmic disk shedding will discharge opsin in the SRS (see Figure 2’s legend) according to their age, preventing the wasting of recently synthesized rhodopsin, the rod protein with the highest expression. Second, rod photoreceptors may use less energy to signal light than rhabdomeric photoreceptors [21,22] (reviewed by [16]). Energy sparing is a critical issue, as the vertebrate eye has a high metabolic rate, with about 80% glucose used for OS renewal rather than for supporting oxidative phosphorylation [23] (recently reviewed in [24], pp. 17–19). Furthermore, during prolonged exposure to intense light stimuli, rods’ membrane potential partially recovers toward its dark level due to the activation of a hyperpolarization-activated current (I_h_) [25,26,27] (reviewed in [28]) and the associated sodium influx may contribute a metabolic load for mammalian rods [21], which are especially sensitive to changes in sodium turnover [29]. 

Using a dedicated enzymatic pathway able to operate 11cis-RAL resynthesis from at-RAL in low light levels did not come without a toll on ciliary photoreceptors. A relevant disadvantage is its intrinsic slowness, which limits photoreceptor dark adaptation after exposure to bright light levels [30]. As shown in Figure 3, photon absorption-triggered cis-trans isomerization eventually leads to at-RAL release from opsin.

The at-RAL released inside the disks of rod photoreceptors must be converted to all-trans-retinol (at-ROL or vitamin A) in the rod cytoplasm by retinol dehydrogenase 8 (RDH8) or 12 (RDH12) [31,32]. The at-ROL is then taken up by interphotoreceptor retinoid-binding proteins (IRBP) and diffuses across the SRS up to the retinal pigment epithelium (RPE) [33]. In the RPE, at-ROL is esterified to retinol ester of a long-chain fatty acid by the enzyme lecithin-retinol-acyl transferase (LRAT) [34] to provide the optimal substrate for the isomerohydrolase activity of RPE65 [14]. RPE65 then converts the retinol ester into 11cis-ROL [14], which is further converted into 11cis-RAL by retinol dehydrogenase 5 (RDH5) [35]. The 11cis-RAL is then taken up by cellular retinaldehyde-binding protein (CRALBP) [36]. IRBP will shuttle 11cis-RAL back to OS by diffusion along the SRS, allowing opsin recovery to the ground state.

It is important to note that the RPE stores retinyl esters. Therefore, the 11cis-RAL synthesis rate may not depend on the enzymatic steps converting at-RAL in at-ROL in photoreceptors and at-ROL diffusion from photoreceptors to RPE, as discussed in [37]. On the other hand, the delays introduced by the serial enzymatic steps in the RPE [30] may add up to the diffusion time across the SRS, with diffusion likely contributing to the delay in rhodopsin recovery to the ground state and the dark-adaptation time (reviewed in [37]).

Considering the 50 µm long SRS in mice [38] and assuming a diffusion coefficient (D) for IRBP-bound 11cis-RAL of 0.5 × 10^−7^ cm^2^ s^−1^, from the relation x=√2Dt, linking the time (*t*) required for a molecule of diffusion coefficient *D* to travel a given space (*x*), the time required for traveling the SRS is 250 s. Considering that the time for full recovery to dark-adapted sensitivity in rods is in the order of several tenths of a minute, the rough estimate above may indicate that in addition to the enzymatic steps of the retinoids cycle, the diffusion time across the SRS also adds up significantly to the overall delay, consistent with the analysis by [37]. Note that the ten-fold difference between IRBP and glucose diffusion coefficients relies on the over 1000-fold difference between IRBP (260 kDa) and glucose (0.18 kDa) MW. However, the viscous hyaluronic acid-based extracellular matrix of the SRS may reduce the IRBP diffusion coefficient in the SRS [39], considering the non-covalent link between the hyaluronan-based matrix and two hyaluronan binding motifs in IRBP [40]. The contribution of 11cis-RAL diffusion along the SRS has been modeled as “resistive” elements in [37], where it was proposed to set the time course of dark adaptation in humans. This modeling is consistent with the notion that shortening the diffusion time would reduce the time required to recover light responsiveness in darkness after photoreceptor adaptation to bright light stimuli.

### Retinal Photoconversion Enables Cones to Operate in Bright Light

In the vertebrates’ duplex retina, cone photoreceptors operate in bright light to extend vision over a light intensity range that may exceed 10 log units [20]. High light intensities and the ensuing high rate of 11cis-RAL isomerization to at-RAL require the fast replacement of at-RAL with newly generated 11cis-RAL. However, the delay imposed by the serial enzymatic steps of the retinoid cycle [30] and the diffusion time along the SRS [37] may fall short of providing such a fast 11cis-RAL turnover. Indeed, 11cis-RAL synthesis may not be limited to the retinoid cycle, and an additional pathway supports cone photoreceptors’ operation in bright light (recently reviewed in [41]). 

The basis for an additional pathway involved in 11cis-RAL synthesis stemmed from the discovery that the retina and RPE express an opsin homolog indicated as the retinal G-protein receptor (RGR) based on its homologies with G-protein-coupled receptors (GPCR) [42], despite RGR lacking the structural motifs required for GPCR interaction with G-protein [43]. RGR’s structural features put it in the group 4 opsin [44] (see also Figure 1 in [41]) that includes the photoisomerase retinochrome expressed in invertebrates [45]; thus, it is unlike either r- or c-opsins. In mice with a targeted disruption of *Rgr* and exposed to continuous bright light level (4000 lux) for 8 h, the decrease in 11cis-RAL and rhodopsin levels was more prominent than in wt mice [46]. In contrast, *Rgr* disruption did not affect 11cis-RAL or rhodopsin in mice dark-adapted overnight. Moreover, following overnight light adaptation, *Rgr* disruption led to an increased accumulation of all-trans retinyl esters (at-RE) compared to wt [46]. These results indicated that *Rgr* codes for a protein contributing to opsin regeneration in light-adapted mice, possibly photoconverting at-RAL to 11cis-RAL. 

However, *Rgr*^−/−^ mice had less than half the isomerohydrolase activity of wt mice, suggesting that RGR may stabilize RPE65 isomerohydrolase activity [47]. RGR has been reported to coprecipitate with RDH5 and RPE65 [48], the isomerohydrolase of the retinoid cycle [13,14], although evidence for a direct interaction has only been found for RDH5 and RPE65 [48]. 

These controversial findings in RPE cells prompted the investigation of RGR’s role in MGCs, which also express *Rgr* [49]. At variance with the RPE, the retina does not express RDH5 [50], indicating that MGCs must express a different RDH. The retina expresses RDH10 [51], which may work with RGR to photoconvert at-RAL in 11cis-ROL. Indeed, the coexpression of *RGR* and *RDH10* in HEK293T cells led to the 11cis-ROL release in the medium in response to a 30-min exposure to 470 nm light [52], while RGR coexpression with RDH5, RDH8, RDH11, or RDH14 did not increase 11cis-ROL release in response to light. A more significant increase in 11cis-ROL production was observed in retinal microsomes derived from wt compared to *Rg*^−/−^ mice [52], while similar 11cis-RAL levels were measured in microsomes from wt and *Rgr*^−/−^ mice, indicating the specificity of 11cis-ROL synthesis in response to light. 

Data from retinal microsomes did not carry insights into RGR expression by a specific retinal cell type. Immunohistochemical evidence of RGR expression by MGCs apical processes, which contact photoreceptor cell bodies at the outer limiting membrane, indicates that 11cis-ROL generated by MGCs may reach photoreceptors via a shorter diffusional path than 11cis-RAL generated by RPE cells [30,52].

Using an ex vivo approach that overcomes the limitation of in vivo ERG recordings, it was found that in isolated retinas (i.e., without RPE cells generating 11cis-RAL) of Gnat1-/- mice (lacking rod responses), Rgr wt cones exposed to a continuous 505 nm high-intensity background light still responded to 565 nm light flashes, clearly demonstrating the occurrence of a pigment regeneration pathway in the retina of wt mice [52]. On the other hand, cones in the isolated retinas of *Rgr*^−/−^ and *Gnat1*^−/−^ mice underwent a faster decrease in response amplitude and sensitivity to light [52], indicating the relevance of the RGR-based pathway in MGCs for opsin regeneration in vivo. RGR-mediated at-RAL photoconversion in 11cis-RAL by human, bovine, and mouse RPE cell microsomes has been reported [53]. The same study, using single-cell RNA sequencing (scRNA-seq), also found evidence for *RGR* selective expression in human RPE and MGCs [53]. Evidence for expression was found in mouse RPE but not in MGCs [53], in contrast to the molecular and functional data in mouse MGCs [52]. Evidence that RGR-based photoconversion of at-RAL requires RGR to bind at-RAL via a Schiff’s base has been provided by the mutation K255A, which abolishes at-RAL binding to RGR [53] and also prevents its photoconversion to 11cis-RAL.

Although these data indicate *RGR* expression by both RPE and MGCs, with MGCs contributing to the photoconversion of at-RAL to 11cis-RAL, an unanswered question was the relative contribution of RPE and MGCs and whether their contribution is redundant. Insights into the relative contribution of RPE and MGCs to 11cis-RAL regeneration have been provided by *Rgr*^−/−^ transgenic mice, generated by the insertion of a stop cassette in an *Rgr* intronic sequence flanked by loxP sequence [54]. By crossing these mice with transgenic mice with tissue-specific tamoxifen-inducible expression of Cre recombinase, *Rgr* expression could be selectively rescued in RPE cells (via *Rpe65* promoter) or in glial/astrocyte cells (via *Slc1a3* promoter) [54]. The functional analysis via ERG recordings in transgenic mice with a *Gnat1*^−/−^ background indicates that RPE and MGCs maintain cone responding when light-adapted, and their contribution is not redundant [54]. The result of scRNA-seq analysis indicates that MGCs could be functionally heterogeneous, as *Rgr* expression was restricted to 21% of MGCs in mice and 5–8% of MGCs in the macaque, a non-human primate [54], suggesting that the difference is not related to the presence a well-defined fovea. Intriguingly, over 90% of human MGCs express *RGR* independently from their foveal or peripheral localization [54]. 

Recent evidence indicates that at-RAL photoconversion by MGCs may play a role in the Alstrom syndrome. The Alstrom syndrome is a genetic disorder that affects multiple organs and leads to early-onset cone/rod dystrophy [55]. In patients, the disease is inherited as an autosomal recessive disorder caused by mutations in *ALMS1* [56]. Mice lacking *Alms1* (*Alms1^−/−^*) also exhibit early cone-rod dystrophy [57]. In these mice, ERG recordings in the isolated retina (i.e., lacking RPE-generated 11cis-RAL) showed an increase in the fast component of cone light-sensitivity recovery after bright light exposure [58], suggesting a role of at-RAL photoconversion by MGCs in this process. These findings highlight the importance of tightly regulating RGR-mediated at-RAL photoconversion by MGCs to maintain cone viability in *Alms1^−/−^* mice.

In conclusion, these findings suggest that RPE and MGCs may play a role in maintaining cone function in bright light through the photoconversion of at-RAL to 11cis-RAL, although the extent of their relative contribution may be species-specific. 

## 3. Retinoids’ Cycle and Photoreceptors’ Viability

Another disadvantage of the retinoid cycle, in addition to its slow speed, is that it may adversely affect photoreceptors’ viability. Like most aldehydes, at-RAL may react in a non-specific manner, forming adducts with the amino group side chain of proteins (reviewed in [59]) and the membrane phospholipid phosphatidylethanolamine (PE). 

As shown in Figure 3, once released from the opsin inside the disk, at-RAL may react with the PE amino group to generate a Schiff’s base N-retinylidene PE (NRPE), a reaction that occurs at a faster rate in native membranes than in reconstituted micelles [60]. Without the flippase transport system coded by *ABCA4* [61], which transfers NRPE to the OS cytoplasm, NRPE will accumulate inside the disk. The rod disk membrane has a higher PE content than cone OS [62]. Nevertheless, *ABCA4* is expressed in both rod [63] and cone [64] photoreceptors, and although ABCA4 may also bind at-RAL and 11cis-RAL, NRPE is its preferred substrate [61]. Upon crossing the disk membrane and entering the OS cytoplasm, NRPE is hydrolyzed to at-RAL and then converted to at-ROL by (RDH8 and RDH12) [31,32,60]. These steps are crucial in clearing at-RAL from the OS [65]. 

Gene variations reducing the effectiveness of these enzymatic steps converting at-RAL to at-ROL have been associated with several retinal degenerations. Table 1 reports data from the database ClinVar accessed on 22 January 2024, for gene variants affecting enzymes/transporters of the retinoid cycle. 

While none of the 9 RDH8 variants are classified as pathogenic or likely pathogenic according to the American College of Medical Genetics and Genomics/Association for Molecular Pathology guidelines [66], for the 506 reported RDH12 variants, 112 have so far been reported as either pathogenic or likely pathogenic, with most cases being clinically classified as Leber Congenital Amaurosis 13 (LCA13). LCA13 is a severe and progressive retinal dystrophy [67,68,69,70] with an altered retinal structure and the loss of cone and rod photoreceptors. For *ABCA4*, ClinVar lists 3252 variants, with 787 and 480 listed as pathogenic or likely pathogenic, respectively. Although initially identified as a gene involved in autosomal recessive Stargardt disease [63] (STGD1), a form of juvenile-onset macular dystrophy, a recent genotype-phenotype correlation matrix indicates that a large number of *ABCA4* variants generate a complex clinical landscape for *ABCA4* disease [71], with a concurrent intronic variant [72] contributing to the most common disease-causing variant in patients of European descent. In addition, there is evidence of different pathogenetic mechanisms in transgenic mice, either knock-out (*Abca^−/−^*) lacking ABCA4 protein or knock-in for a gene coding a misfolded ABCA4 protein, indicating that the lack of enzymatic activity may not fully account for the clinical presentation in patients [73]. 

These data indicate that the basis for the clinical variability of retinal dystrophies associated with gene variants involved in at-RAL detoxification is still being defined.

### 3.1. Mechanisms Linking Retinoid Cycle Defects to Impaired Photoreceptors’ Viability

It is important to note that uncertainties also exist regarding the mechanisms responsible for the damage induced by at-RAL derivatives. As individuals age, lipofuscin deposits begin to accumulate in the RPE [74]. Lipofuscin deposits contain bisretinoids, such as retinylidene-N-retinyl ethanolamine (A2E) [75,76]. A2E results from the reaction of a second at-RAL with NRPE to generate A2PE [75,76], which is hydrolyzed to A2E in the RPE [75,76]. 

Upon A2E’s exposure to light, the mixture of A2E isomers in the RPE [76] indicates that retina-generated A2PE may be taken up by RPE as a result of disk-shedding and undergo further conversion in the RPE in response to light. Furthermore, an additional condensation product generated by the reaction of at-RAL dimers with PE was identified in bleached OS in vitro and RPE isolated from human donor eyes [77]. An intriguing property of these at-RAL derivatives is their photo-oxidation with the addition of oxygen at double bonds, generating reactive oxygen species [77]. The finding of decreased A2E derivatives in mice with reduced synthesis of 11cis-RAL by RPE cells [78] is most likely due to the reduced at-RAL release from opsins and the decreased A2PE generation that will suppress A2E build-up in RPE cells. 

Evidence suggests that at-RAL condensation products (bisretinoids) may act as photosensitizers that trigger oxidative stress in RPE and retinal cells in response to light (recently reviewed in [79]). 

However, the sequence of events leading to A2E accumulation in the RPE and A2E photo-oxidation as the only cause of retina damage is uncertain. The first point is the higher PE levels in the OS of rods than in those of cones [62], which explains the higher NRPE levels in rods than in cones [80]. Despite this finding, pathogenic *ABCA4* variants were initially associated with autosomal recessive Stargardt 1 disease, a juvenile-onset macular retinopathy [51], i.e., primarily affecting cones. 

Differences between rods and cones in the rate of A2PE synthesis as a function of the at-RAL to PE ratio (at-RAL:PE) have been addressed in synthetic membranes (egg PE) lacking NRPE clearing by ABCA4 [81]. Decreasing the at-RAL:PE from 4:1 to 4:8 led to a pronounced increase in the rate of A2PE generation [81], suggesting that increasing PE availability for each at-RAL will increase A2PE synthesis. On the other hand, A2PE generation increases up to a 4:2 at-RAL/PE and plateaus with an 8:4 ratio [81] (recently revised in [82]), perhaps due to the first retinyl moiety bound restricting the access of a second at-RAL to the reactive nitrogen by steric hindrance. These results suggest that increasing at-RAL:PE by reducing PE suppresses the rate of A2PE generation in cones. However, this conclusion is hard to reconcile with the notion that cones have a higher liability than rods to the loss of ABCA4 activity despite their lower PE content. 

Indeed, rods’ higher PE [62] would promote their A2PE accumulation compared to cones. However, caution is required when translating the results from artificial membranes to native OS. The A2PE generation rate was higher with egg PE than with docosahexaenoic acid (DHA)-containing PE, indicating PE lipids affect A2PE synthesis. This finding is relevant as rods have higher DHA content than cone OS [62], suggesting that the increased DHA-containing PE may suppress the rate of A2PE generation in rods despite their higher PE content than cones.

Several transgenic mouse models have been available to investigate the role of specific genes in retinal dystrophies, and Table 2 reports gene function with the mouse phenotype.

The difference between rods and cones in their liability to pathogenic *ABCA4* variants has also been explored using Neural Retina Leucine zipper KO mice (*Nrl^−/−^* mice). These mice, lacking a gene critical for the maturation of rod precursors [83], were used as a pure-cone retina model because rod-fated photoreceptors appear to convert into blue cones [84,85]. However, A2E levels in the RPE of *Abca4^−/−^* mice were higher in the rod-dominant (*Nrl*^+/+^) (rods make up 97% of total photoreceptors in the mouse retina [86]) than in the cone-dominant *(Nrl*^−/−^) retinas (see Figure 8B in [80]), and the difference became significant in mice raised for 120 days in bright (300 lux) cyclic light (Figure 7 in [80]), i.e., in conditions promoting at-RAL generation from 11cis-RAL. Moreover, RPE cells’ lipofuscin granule number and size were higher in the RPE of rod-dominant than in the cone-dominant retinas of *Abca4^−/−^* mice (see Figure 9 in [80]). Despite absolute values for A2E content and lipofuscin granules being higher in the RPE of rod-dominant than in the *Abca4^−/−^ Nrl^−/−^* double knock-out mice (pure-cone retina lacking ABCA4), the RPE of *Abca4^−/−^ Nrl^−/−^* double knock-out mice contained a 6.8-fold higher A2E per pmole of 11cis-RAL than in *Abca4^−/−^* mice (rod-dominant retina with non-functional ABCA4) [80]. 

These results suggest an increased A2E generation in RPE cells facing cones compared to rod cells when the amount of opsin is considered. However, after accounting for the 40% reduction in the photoreceptor number [84] and the 75% reduction in the OS volume in *Nrl^−/−^* compared to wt mice (see Table 1 in [87]), similar opsin content per OS unit volume has been estimated in rods and *Nrl^−/−^* cones [84,87]. 

Despite similar opsin content for unit OS volume, it could be argued that cones contribute more at-RAL for each opsin molecule due to rod saturation at high luminance levels. However, rod saturation indicates that rod cGMP-gated channels are closed, not that all rhodopsin molecules have been isomerized. This point has been demonstrated at both the retinal and behavioral levels by showing that mouse rods recovered their light responsiveness within 20–30 min of continuous light exposure to an adapting light that initially drove them into saturation [88]. Of note, light-adapted rods respond to light intensities (up to 10^7^ photoisomerizations rod^−1^ s^−1^) where only cones were thought to operate [88]. The underlying mechanisms involve bleaching adaptation and the slow translocation of transducin from the OS to the IS [89]. Intriguingly, this mechanism involves blue-light-induced 11cis-RAL regeneration, a finding consistent with data showing that RGR-mediated at-RAL photoconversion in the RPE and MGCs contributes to both scotopic and photopic vision [54]. 

As a result, after accounting for differences between photoreceptor number and OS volumes and for rods’ ability to operate over a range of light intensities that are superimposed with the one where only cones were formerly thought to operate, the amount of A2E generated in the RPE may be similar in rods and cones. In this regard, the observation that cones demise in the *Nrl^−/−^ Abca4^−/−^* double-mutant mice did not proceed faster than in *Nrl^−/−^* led the authors to consider that this murine model may not fully recapitulate the events occurring in human foveal cones [80]. 

To avoid the limitations of animal models lacking foveal cones, differences in the lipofuscin accumulation between health controls and Stargardt 1 patients were found by monitoring the eye fundus fluorescence [90]. A confocal scanning laser ophthalmoscope (cSLO) equipped with an internal fluorescence reference allows the quantification of fundus autofluorescence (qAF) differences between foveal and extrafoveal regions [91]. Although lipofuscin contains several bisretinoids, A2E represents the best-characterized component, and qAF may provide a proxy for A2E accumulation [90] reflected by lipofuscin fluorophores in RPE. 

In healthy subjects, qAF increases with the subjects’ age, and the spatial profile shows the highest values superotemporally with the minimum intensity at the fovea [92]. The foveal qAF minimum depends on the excitation wavelength, with lower values measured using 488 nm light than when using 550 nm [92]. The reduced signal with 488 nm excitation light may reflect its filtering by the macular pigment with peak absorbance at 460 nm [93]. Measurements of A2E and other bisretinoids in the human retina confirmed that higher A2E levels were present in the periphery than in the macula [94], and an overall A2E increase was observed in aged donors [94]. Healthy control and Stardgardt patients had similar qAF spatial profiles but higher fluorescence in patients’ foveal and extrafoveal regions than healthy controls [95]. 

A linear correlation between fundus qAF and A2E content in RPE was found in wt and *Abca4^−/−^* mice [96]. However, the relation between the qAF and A2E contents broke down with aging in mutant mice due to decreased A2E, despite increasing qAF [96]. The decrease in A2E content correlates with the reduction in ONL thickness [96], a finding that may indicate A2E levels in the RPE represent the balance between the synthesis of A2E’s precursor, A2PE, in rod photoreceptors and A2E conversion to other fluorescent compounds that accumulate in RPE lipofuscin granules. 

In humans, the lower A2E levels and qAF in the foveal region may also mirror a balance between A2E synthesis from A2PE generated in photoreceptors and A2E photodegradation in response to blue light (400–490 nm) [97,98], which causes A2E photodegradation [99] in A2E-laden RPE cells. Blue light exposure generated several toxic compounds [100,101,102,103,104,105] in human RPE cells [99,100,101]. In particular, the generation of reactive peroxides [102] and other oxidized [103] compounds in murine and human RPE cells in response to blue light suggests that A2E photodegradation in the macula may cause RPE oxidative stress and damage. Indeed, A2E levels were similar in albino and pigmented mice reared in darkness, but they decreased significantly by 45% in albino mice housed in cyclic light. In contrast, no significant decrease was observed between different genotypes in pigmented animals [97]. In albino mice reared in cyclic light, the oxidized A2E does not accumulate, indicating it is further degraded [97]. 

In *Abca4* KO albino mice (*Abca4^−/−^*), vitamin E treatment for five months significantly and dose-dependently suppressed the differences in the content of bisretinoids between dark- and cyclic light-reared mice [97]. Intriguingly, suppression by vitamin E of bisretinoid photodegradation correlates with the suppression of ONL thinning, suggesting that photodegradation products may play a role in photoreceptor damage. The finding of short-chain aldehyde generation in response to blue light exposure (2 h a day for seven days) [97] and the accumulation of methylglyoxal adducts in proteins indicate possible mechanisms linking A2E photodegradation with cellular damage [102].

Recent evidence indicates that increased plasma iron levels are associated with an enhanced generation rate of highly reactive hydroxyl radicals (OH.) [79,103]. An increased level of intracellular iron also leads to reactive species generation and A2E degradation [104], suggesting that A2E oxidation may contribute to iron toxicity. 

A high-fat diet is an additional environmental factor that increases retinal PE levels and A2E synthesis [105]. However, these effects were observed in mice kept on the high-fat diet for 3, 6, and 12 months, i.e., for a substantial part of a mouse’s lifetime, and it is unclear whether they may relate to a juvenile-onset maculopathy. 

Data on the generation and degradation of bisretinoids by RPE cells exposed to blue light may provide insights on foveal cone degeneration preceding that of rods in Startgardt 1 patients. The links between blue light focusing on the fovea, bisretinoid oxidation, and ONL thinning may also explain foveal cone damage in age-related macular degeneration (AMD) patients. However, considering the 60 diopters (60D) of an emmetropic eye and the 1D difference between 550 and 450 nm light due to axial chromatic aberration [106], the 450 nm blue light is expected to focus about 280 µm before the 550 nm light (see also Figure 4 in [107]). Consequently, blue light energy will spread over a larger area at the RPE than the retinal level, covering an RPE area exceeding the foveal pit [108,109,110]. Moreover, the 1D axial chromatic aberration will focus blue light close to the macular carotenoids of cone axons (Henle fibers), attenuating blue light energy [111,112]. Interestingly, macular pigment lutein and zeaxanthin have been reported to reduce the probability of progressing from intermediate to advanced AMD when substituting β-carotene in the AREDS2 formula [113], although their protective role is disputed [114].

Another point that deserves attention is the role of vitamin E in suppressing ONL thinning, which suggests a role for oxidative damage in blue light-induced foveal cone demise. Several studies have addressed the protective role of vitamin E in AMD patients. An initial prospective placebo-controlled randomized control study in 1193 healthy subjects aged 55–80 who received 500 mg of vitamin E daily for four years [115] found a similar incidence for early AMD (8.6% vs. 8.1% for vitamin E vs. placebo, respectively, with a relative risk of 1.05 and CI of 0.69–1.61) as well its progression to advanced forms (0.8% vs. 0.6% for vitamin E vs. placebo, respectively, with a relative risk of 1.36 and CI of 0.67–2.77). 

A systematic review and meta-analysis on a total of 23,099 people, randomized in three trials with a treatment duration of 4–12 years, found no evidence that antioxidant (vitamin E or *β*-carotene) supplementation prevented AMD (the pooled risk ratio was 1.03 (95% CI, 0.74–1.43)) [116]. 

On the other hand, in people with signs of the disease, the AREDS formula combining antioxidants (*β*-carotene, vitamin C, and vitamin E) and minerals (zinc and copper) slowed down the progression from intermediate to advanced AMD and visual acuity loss (adjusted odds ratio = 0.68, 95% CI, 0.53–0.87 and 0.77, 95% CI, 0.62–0.96, respectively) [116]). Of note, macular pigments lutein and zeaxanthin, besides their antioxidant properties [113], may afford additional protection to cones by dynamic filtering blue light impinging on the macula [117], while improving contrast luminance detection and reducing glare in healthy human subjects [118] (reviewed in [107]). 

The Blue Mountains Eye Study, testing 2454 subjects at several intervals over ten years, found that vitamin E increased the relative risk of advanced AMD (RR = 2.83, 95% CI 1.28–6.23) in subjects in the upper third of intakes compared to the lower third [119]. 

An additional large RCT did not provide evidence that vitamin E at large doses (600 mg) on alternate days vs placebo for ten years in 39,876 women older than 45 years prevents early AMD (117 vs. 128 for vitamin E vs. placebo, respectively, with a relative risk of 0.93 and CI of 0.72–1.19) [120]. 

These results suggest that vitamin E may not protect from early AMD but will delay the progression from intermediate to advanced forms when administered with other antioxidants and minerals. These findings are surprising considering that quenching singlet oxygen by vitamin E prevents A2E oxidation in response to blue light exposure and the generation of glyoxal and other reactive aldehydes, which may form protein adducts [97,102]. 

The reduction in photoreceptor viability is not restricted to gene variants impairing at-RAL detoxification, as gene variants impairing 11cis-RAL synthesis also reduce photoreceptor viability. For instance, pathogenic or likely pathogenic *LRAT* [121,122,123] or *RPE65* [124,125] (recently reviewed in [126]) variants may lead to the early loss of cones, preceding that of rod photoreceptors, even though these patients had reduced bisretinoid accumulation [127,128]. Retinal dystrophies associated with pathogenic variants impairing the conversion of retinyl esters into 11cis-RAL have been attributed to unliganded opsins in the OS. Indeed, supplementation with 9cis-RAL, which leads to the formation of iso-rhodopsin, has been found to increase the dark-adaptation rate and cone response [129] in a mouse model of retinal dystrophy (Fundus albipunctatus) associated with a lack of dehydrogenases activity in RPE cells [35]. Furthermore, cone loss in mice lacking either LRAT or RPE65 activity was partially prevented by administering 9cis-retinyl acetate, a pro-drug for 9cis-RAL [130]. In a non-randomized prospective phase I trial, seven patients diagnosed with Fundus albipunctatus receiving 9cis beta-carotene showed a significantly improved rate of their rod response and an improvement in visual field amplitude [131]. These findings confirm that photoreceptor loss may result from unliganded opsin and photoreceptor rescue by 9cis-RAL supplementation via iso-rhodopsin formation.

### 3.2. Dysregulated Lipid Metabolism in RPE and Photoreceptors’ Demise 

*PROM1* codes for the transmembrane protein Prominin-1 (alias CD133), expressed by stem cells, photoreceptors, and RPE cells. CD133 may have different functions in proliferating and quiescent cells [132]. Prominin-1 may control stem cell activation by recruiting ciliary components and coordinating with sonic hedgehog (SHH) to convert quiescent stem cells into proliferating transient amplifying cells [133] or organizing the cilium required for OS morphogenesis in photoreceptors [134,135] and regulate autophagy in RPE cells by controlling autophagosomes maturation and trafficking to lysosomes [136]. However, Prominin-1 is also a cholesterol-binding protein promoting axonal regeneration by down-regulating neuron cholesterol synthesis via the Smad pathway [137]. It is intriguing that, although not involved in the retinoid cycle, *PROM1* pathogenic variants in Stargardt 4 patients may cause autosomal dominant macular degeneration with increased macular fundus autofluorescence [138]. 

Similar considerations may hold for other gene variants causing retinal dystrophy with macular degeneration. A2PE generated from at-RAL in photoreceptors will reach the RPE via OS phagocytosis and be converted into A2E. This notion is consistent with the impact of pathogenic gene variants on the retinoid cycle, such as *ABCA4*. However, *ABCA4* expression in the RPE may indicate that the variant impact in Startgardt 1 patients may not be restricted to suppressing A2PE synthesis in photoreceptors and, like *PROM1* variants, may result from defective lipid handling by RPE cells. 

Indeed, mouse and human RPE cells express *Abca4*, although at about 1% of its retinal level [139]. Despite *Abca4* expression in RPE cells being lower than in photoreceptors, transgenic mice expressing *Abca4* in the RPE but not in photoreceptors show reduced damages compared to *Abca4^−/−^* mice [139], along with the significant reduction in the content of bisretinoids and lipofuscin granules in RPE cell cells. Intriguingly, Abca4 immunolabelling in RPE cells indicates its colocalizations with endolysosomal markers [139], suggesting faulty phagosome processing and trafficking to lysosomes as a possible common link between Stardgardt 1 and 4 retinal dystrophies.

Independent evidence for the cell-autonomous role of the RPE in retinoids and lipid processing results from the analysis of retina-naïve RPE cells either derived from human inducible stem cells (hiPSCs) lacking *ABCA4* (*ABCA4^−/−^*) or from a Stargardt 1 patient. In *ABCA4^−/−^* hiPSC-derived RPE cells fed for seven days with wt OS, lipid accumulation was up to 2- to 3-fold higher than in control hiPSC-derived RPE cells. This finding indicates the impaired degradation of wt OS by faulty RPE cells [140]. The analysis of labeled wt POS at two different times indicates their normal trafficking to lysosomes. However, the increased lysosomal retention time points to the defective digestion of wt POS by lysosomes of RPE cells derived from *ABCA4^−/−^* hiPSCs [140], indicating that the lack of functional ABCA4 impairs OS processing by RPE lysosomes. Lysosomal alkalinization in RPE cells of *ABCA4^−/−^* mice and human RPE cell line ARPE-19 exposed to bisretinoids has been associated with defective OS processing [141]. In *ABCA4^−/−^* hiPSC-derived RPE cells, reducing lysosomal alkalinization partially rescued wt OS digestion [140], suggesting that loss of ABCA4 function interferes with the control of lysosomal pH. 

In mice, the selective loss in the RPE of *ABCA1*, coding for the lipid transporter ABCA1, causes lipid accumulation in the RPE, retinal inflammation, and RPE and retina degeneration [142], indicating the crucial role of lipid handling by RPE cells, for their own and retinal viability. In *ABCA4^−/−^* hiPSC-defective RPE, the upregulation of *ABCA1* expression in response to the activation of the Liver X Receptors (LXR) reduces lipid deposits compared to *ABCA4^−/−^* hiPSC-derived RPE cells not exposed to the LXR agonist [140]. Although these data do not prove that ABCA4 operates as a lipid transporter, they indicate that the lack of ABCA4 may impair lipid homeostasis in RPE cells, independently of its effects on at-RAL detoxification in the OS. 

The notion of dysregulated lipid homeostasis in *ABCA4^−/−^* hiPSC-derived RPE is consistent with the increased cholesterol content and ceramide levels in the apical membrane of *ABCA4^−/−^* hiPSC-derived RPE cells [140]. In RPE cells of *Abca4^−/−^* mice, increased cholesterol has been found to promote apical membrane ceramide accumulation [143]. In turn, the accumulation of ceramides in the apical membrane of RPE cells has been linked to the expansion of early endosomes [144], causing the entry of the complement component C3, whose cleavage into C3a eventually activates the innate immune response [145,146,147,148,149]. In addition, the expansion of early endosomes triggers the activation of the mechanistic target of rapamycin (mTOR) [144], which may lead to the metabolic reprogramming of RPE cells.

The notion that the dysregulation of lipid handling may play a critical role in RPE cell identity and viability is underscored by the evidence that polyunsaturated lipid peroxisomal β-oxidation is required for lysosome processing of phagocyted OS and for preventing RPE cell dedifferentiation [150]. Peroxisomal β-oxidation is involved in docosahexaenoic acid (DHA) synthesis [151,152,153] and very long polyunsaturated fatty acid metabolism [154], enriched in rods compared to cones [62,155,156]. This difference may have a role in cones having higher sensitivity than rods toward defects in the trafficking and processing of phagocyted OS. Indeed, compared to the *Abca4^−/−^* mouse model, where the retinal degeneration became appreciable eight months after birth, early photoreceptor demise starts at eight weeks in mice with peroxisomal β-oxidation deficiency in the RPE [150,157]. DHA supplementation appears to delay the early degeneration of photoreceptors in mice with peroxisomal β-oxidation deficiency without blocking it [157], indicating that DHA shortage plays a role. Still, additional factors may become limiting in mice fed with DHA-supplemented formula. 

An intriguing aspect of the mixed impact of DHA on RPE cell survival is that several prospective studies (reviewed in [158]) found people with higher intakes of polyunsaturated n-3 fatty acids, such as DHA and eicosapentaenoic acid (EPA), have a reduced risk of developing early AMD, but DHA and EPA supplementation did not prevent AMD from progressing from an intermediate to an advanced form [159]. 

This evidence indicating that genetic defects thought specific for photoreceptors instead also affect RPE cells’ function, identity, and viability points to the RPE cells as a prominent player in retinal dystrophies. 

Serial analysis of eye fundus in STGD1-diagnosed patients using near-infrared (NIR) autofluorescence (AF) to monitor melanin as a proxy for RPE cells and blue short-wavelength (blue SW-AF) autofluorescence to monitor bisretinoid autofluorescence over a relatively narrow field (30°–50°) of the macular region as a proxy for cone photoreceptors where the bisretinoids were generated [160] indicate that the loss of the NIR-AF signal precedes the formation of bright spots (flecks) in blue SW-AF. These results, suggesting that the macular increase in bisretinoid autofluorescence follows the RPE damage [160], were confirmed over larger macular areas (200°) using a medium wavelength (green MW-AF) [161]. These findings suggest that the increase in bisretinoid correlates with retinal damage but appears as the consequence rather than the cause of RPE defects, which may lead to cone demise due to faulty metabolism.

The link between bisretinoids triggering lysosome defects and RPE cells’ metabolic reprogramming and dedifferentiation is intriguing, considering the evidence showing that photoreceptors depend on the RPE cells for their energetic metabolism. As mentioned in Section 2, rod photoreceptors require less energy to signal increased light levels than rhabdomeric photoreceptors [16,21,22]. Compared to rods, cones are far less energy efficient [87], and rods support cones’ energetic metabolism by releasing the rod-derived cone viability factor (RdCVF), a diffusible factor that promotes glucose uptake by cones [162]. This notion is part of the concept of the metabolic adaptation to aerobic glycolysis of rods, cones, and RPE cells that partake in what has been dubbed a “metabolic ecosystem” [163] (reviewed in [24], Section 3.2). A critical feature of the metabolic ecosystem is that RPE cells must spare glucose for photoreceptors, which use 80% of their glucose supply for lipid synthesis [23], to build new plasma membranes required to replace those shed daily with the OS. In turn, RPE cells receive lactate from photoreceptors and use it for their oxidative metabolism [163]. A critical part of this arrangement is the expression of the glucose transporter GLUT1 (coded by *Slca2a1*). Mosaic deletion of *Slca2a1* in RPE cells [164] or photoreceptors [165] causes retinal patches with OS shortening and photoreceptor loss. When considering the importance of the metabolic specialization of RPE cells for photoreceptor viability, it is relevant to note that the metabolic requirements of the primate retina as a function of eccentricity have been estimated to peak in the fovea, suggesting that the metabolic requirements of foveal cones substantially exceed those of rods [166]. The higher metabolic requirement of cones than rods may contribute to their early demise in response to RPE cell reprogramming that follows dysregulated lipid homeostasis. Interestingly, *Slca2a1* expression is regulated during RPE differentiation with a temporal profile similar to those of genes coding for enzymes of the retinoid cycle, such as *Rpe65* [167], i.e., synchronized with the development of the light response. 

## 4. Photoreceptor and RPE Cells Development and the Regulation of the Retinoid Cycle

The evidence in Section 3 suggests the need for careful control of retinoid flux between the RPE, MGCs, and photoreceptors, as either an increased or a reduced 11cis-RAL synthesis may adversely affect photoreceptors’ viability. Furthermore, recent evidence indicating that reducing 11cis-RAL synthesis may improve cone viability when rods have degenerated [58] suggests cone loss may stem from an imbalance between 11cis-RAL synthesis and the amount of unliganded opsin available to bind 11cis-RAL. 

In adult animals, the concentration gradient driving IRBP-bound 11cis-RAL diffusion along the SRS [37] requires a balance between 11cis-RAL synthesis by RPE cells and opsin expression by photoreceptors. Indeed, 11cis-RAL binding by cone opsins and rhodopsin may act like a sink, generating the concentration gradient between RPE and photoreceptors [37]. It is intriguing to note that during retinal development, RPE cell generation precedes cone and rod photoreceptors’ birthdates and the development of their OS, where opsins accumulate [168,169,170]. A lack of coordination between 11cis-RAL synthesis by RPE cells and opsin expression during development may overcome opsin binding sites, adversely affecting cone photoreceptors. The coordination between 11cis-RAL synthesis by the RPE and opsin expression has not been addressed experimentally nor discussed in a review. To fill this gap, we will discuss the evidence on 11cis-RAL synthesis regulation by the RPE and opsin expression by photoreceptors during adulthood and human eye development. We will then address the possible role of at-ROL 9cis derivatives in coordinating 11cis-RAL synthesis with opsin expression.

### 4.1. Development of Retinal Cells Involved in 11cis-RAL Synthesis and Isomerization

The establishment of the presumptive retina starts within the foremost region of the neural plate. In humans, following neural tube closure by the end of the fourth fetal week (FW4) [171,172], the optic vesicles evaginate, start to undergo enlargement [173], and are gradually surrounded by mesenchymal cells, except at the apex, where they remain in close contact with the lateral surface ectoderm. As shown in Figure 4a, at this stage, a disk-shaped thickening of the neural ectoderm, known as the retinal disk and corresponding to the developing neural retina, resides beneath a localized thickening of the surface ectoderm, recognized as the lens placode [174,175]. 

In the following developmental stages (Figure 4b), the optic vesicle forms a single-layered sphere that then invaginates to create the two layers of the optic cup. The inner layer will develop into the neural retina, while the outer layer will become the retinal pigment epithelium (Figure 4c). Specifically, by the eighth week of fetal development (FW8), the retina begins to differentiate into a thin outer layer that will become the retinal pigment epithelium (Figure 4d), separated by a narrow subretinal space from a thicker inner neural retina [176]. A differentiation wave starts near the optic stalk and moves toward the periphery, creating inner and outer neuroblastic layers in the neural retina (Figure 4d) [177]. Simultaneously, mesenchyme condenses around the outer surface of the optic cup. The innermost layer of this mesenchyme, which is loose and highly vascular, will become the choroid, while the outer layer will form the sclera [178]. 

The gradient of retinal differentiation determines the different stages of retinogenesis along the central–peripheral axis. Retinal progenitor cells give rise to all types of retinal cells in a coordinated manner. Retinal ganglion cells, horizontal cells, and cones are generated during the first wave, followed by a second wave in which rods, bipolar, and MGCs are born (Figure 5) [170]. Amacrine cell generation bridges the temporal gap between these two phases [179,180]. Given the significance of differentiation and interactions among photoreceptors, the RPE, and MGCs in the retinol functional cycle, we delve into a detailed analysis of the development of these cell types.

The retina is a complex structure with multiple layers and various cell types. The RPE is a monolayer of cells connected to the choroid membrane through Bruch’s membrane.

#### 4.1.1. Photoreceptors

Photoreceptor differentiation unfolds across the retina, progressing from the central fovea to the periphery. Cones first appear in the prospective fovea at FW 10.5–11 (Figure 4d) in a pure cone area (PCA) with all five cell layers [169]. At this stage, rods can be found at the edge of the PCA and extend in the periphery up to the occurrence of an outer plexiform layer [169]. At midgestational age (FW 20), the ONL of the PCA edge has a layer of cone and one of the rod nuclei with an IS and no clear OS [169]. By FW 34, rods and cones near the PCA edge have elongated OS [169], indicating that OS maturation lags cell generation by about 13 weeks. It is interesting to note that photoreceptor specification assessed by rhodopsin mRNA expression starts at the edge of the PCA by FW 15, and most rods express rhodopsin mRNA by FW 18 [169], indicating mRNA expression lags rod cell generation by about five weeks. Rhodopsin immunoreactivity starts at about FW16 on the edge of the PCA. Still, in the absence of OS, the immunoreactivity is partially found in the IS cytoplasm and cell membrane and will stay there for 1–2 months [169] before being restricted to the OS. Notably, at birth, some rods in the retinal periphery have yet to express rhodopsin, indicating ongoing rod generation and maturation postnatally. These data indicate that the OS-restricted opsin labeling typical of adult rods may require several months following the rod birthdate.

The genetic pathway driving multipotent progenitors to adopt a photoreceptor fate is well characterized by the initial expression of PAX6 and the subsequent repression of NOTCH, leading to OTX2 expression. OTX2 is a transcription factor crucial for photoreceptor fate specification by activating factors such as VSX2, PRDM, and CRX [181]. After acquiring a photoreceptor fate, cells undergo the decision-making process to differentiate into either cones or rods. The transcription factor NRL plays a pivotal role as a determinant of rod fate [83]. NRL accomplishes this function by a dual mechanism: repressing the expression of thyroid hormone receptor beta (*THRB*) and S-opsin, critical genes for cone specification, and activating nuclear receptor subfamily 2 group E member 3 (NR2E3). NR2E3, in turn, suppresses cone fate and activates a specific subset of rod genes, including *RHO* [182]. Cones are determined by the concomitant expression of OTX2 and the transcription factor ONECUT, which lead to activation of the nuclear thyroid hormone receptor THRB/B2 [183,184]. Thyroid hormone regulators govern the generation of cone subtypes, requiring reduced hormone signaling for the specification of S cones and increased signaling for the specification of M/L cones [75]. Cones committed to the M/L cone fate will later differentiate into M or L cones. The genetic mechanisms guiding this decision are still a subject of investigation, with several proposed models. Thus far, the only clearly recognized difference between M and L cones is the expression of their respective opsin photopigments. Compared to other non-human primates, a distinctive human feature is the remarkable variability in the ratio of L to M cones. This variability suggests biological plasticity, as individuals with extreme L:M ratios still maintain normal color vision [185]. During the generation of photoreceptors, they undergo local patterning, leading to a precise spatial mosaic arrangement of rods and cones in the adult retina, which becomes well organized into longitudinal columns by FW 21 [186]. In particular, L and M cones achieve their peak spatial density at the center of the fovea [187], while S cones are absent from the central 100 µm [187], and rods are absent from the central 300 µm of the adult fovea. Beyond the foveal center, all four photoreceptor types are present, with rods reaching their peak density in a region close but eccentric to the optic disc. Outside the central retina, rods significantly outnumber cones, with a ratio of 1:20 [188]. While the creation and maintenance of this photoreceptor mosaic are essential for the optimal functioning of the human visual system, the exact molecular and cellular processes responsible for its formation remain poorly understood.

Although the intrinsic molecular events underlying rod and cone photoreceptor specification have been identified, the control over OS elongation and opsin sorting to the OS may depend on extrinsic factors (reviewed in [189]), as previously reported for low molecular weight diffusible factors [190,191], such as taurine [192] and the at-ROL derivative retinoic acid [193]. In addition to local diffusible factors, the increase in tensile strength from the inner to the outer retina [194,195], driven by the increase in the number of cells for unit volume (cellular density), may also play a role during a critical period. Recent evidence indicates that a reduced cellular density may result in hybrid rod precursors, which maintain opsin expression but fail to switch off the expression of genes of the MGCs’ fate [196], indicating the importance of local factors linked to cellular density in photoreceptor specification. Chemical gradients of glucose and oxygen partial pressure (pO_2_) along the SRS represent additional environmental factors controlling the expression of genes involved in aerobic glycolysis (recently reviewed in [24]), the metabolic adaptation required to support OS generation and renewal. In the rhesus monkey, a seven-day-long distancing of the retina from the choroid by 100 µm led to a loss of OS, while IS appeared preserved [197]. Retinal reattachment promoted OS growth after the seven-day-long detachment. Still, the rods and cones’ average lengths were half their control values after 30 days, and even after 150 days after reattachment, the cone OS length did not reach the control value [197]. Furthermore, after 30 days of reattachment, the OS growth rate was about 2/3 of the control, suggesting that the limited growth may not simply result from the reduced glucose and oxygen diffusion along the SRS [197]. These results may instead indicate that a relatively long time is required to fine-tune the transcriptional changes underlying photoreceptors’ metabolic adaptation needed for OS growth [198]. 

The notion that the long delay between photoreceptors’ birth and opsin incorporation into their OS is dictated by the time required to develop a metabolic adaptation to environmental cues such as glucose and pO_2_ postpones photoreceptors’ maturation well in the postnatal period. This notion is consistent with the functional analysis of the light response by ERG recording in infants 5–270 days old [199]. By applying the analysis initially developed for an isolated rod photoresponse to the a-wave leading edge of the ERG (reviewed in [37]), it turned out that the overall amplification constant of the phototransduction cascade was substantially reduced in children, reaching the adult value after 3–4 years from birth. A reduced opsin concentration long after birth may play a role in the lower amplification constant, consistent with morphological and molecular data, although the contribution to the reduced amplification constant of geometrical factors, such as differences in eye size and cytoplasmic volume, may complicate the interpretation. 

#### 4.1.2. Müller Glia Cells

MGCs are unique among glial cell types, as they are among the last generated during retinogenesis from progenitors within the retina.

Shortly after their birth, occurring approximately at FW 12, as indicated by the stem cell marker Nestin expression, MGCs actively shape nascent circuits [200]. This involvement coincides with the growth of their morphological complexity and the release of molecules crucial for neuronal survival and synaptogenesis. MGCs, distinctive radial glial cells, extend across the entire retina thickness, making contact with and ensheathing all retinal neurons. Upon their generation, MGCs play a pivotal role in promptly establishing and maintaining the retinal architecture [201]. Furthermore, MGCs are a crucial anatomical bridge connecting retinal neurons with essential compartments for molecular exchange, including retinal blood vessels, the vitreous body, and the subretinal space. The latter forms the pathway to choroidal blood vessels with the RPE. Beyond its anatomical role, this link is also functional due to the rich array of ion channels, ligand receptors, transmembrane transporter molecules, and enzymes found within MGCs and to the contribution to the retinoid photoconversion that supports cone vision in bright light [52,54], as mentioned in Section Retinal Photoconversion Enables Cones to Operate in Bright Light. By FW 18, the appearance of the inner limiting membrane, a basal membrane synthesized by the basal foot processes of the MGCs, denotes the conclusion of their morphogenesis and organization. 

MGCs support neuronal survival and the facilitation of regular information processing. In mice, recent data indicate that the improved viability of peripheral cones after rods have degenerated involves all-trans-retinoic acid (at-RA) signaling in MGCs [202]. Interestingly, MGCs in the central retina have lower levels than the peripheral retina of *Aldh1a1* [202], the gene coding for the aldehyde dehydrogenase converting at-RAL into at-RA. Upregulating Aldha1 in MGCs of the central retina increased cone survival after rods had degenerated [202], indicating that, in addition to 11cis-RAL photoconversion [52], improved cone viability represents a novel role of MGCs.

#### 4.1.3. Retinal Pigment Epithelium

RPE cells, like epithelial cells, are polarized. On the apical side, the RPE extends microvilli, which are long, thin cellular membrane protrusions that intimately contact and surround the outer segments of photoreceptors and facilitate their daily renewal through phagocytosis [203]. The basal side of the RPE has basal infoldings and short invaginations that effectively augment the surface area of the RPE. The RPE gradually establishes a connection with the anterior layer of Bruch’s membrane, an integral component of the choroid consisting of a five-layer extracellular matrix. Furthermore, the polarized distribution of ion channels and transporters within the RPE regulates the composition of the subretinal space. This regulation, in turn, supports the survival and proper functioning of photoreceptors and other types of retinal cells. 

Human RPE cells begin expressing melanin at FW 5, rendering this tissue visible on the external examination of the embryo. At later stages, the RPE is a monolayer of tightly adherent cuboidal cells characterized by central round nuclei. The RPE cell fate is crucially controlled by transcription factors MITF and OTX2 [204]. In murine models, *Mitf* inactivation leads to retina generation instead of RPE. Conversely, the ectopic *Mitf* retina expression inhibits neurogenesis and induces the expression of genes associated with pigment biogenesis [205,206].

### 4.2. Regulation of the Retinoid Cycle

Control over 11cis-RAL synthesis may involve transcriptional and posttranscriptional mechanisms in adult RPE cells. Considering the sequential steps involved in the 11cis-RAL synthesis, the expression of genes such as *LRAT*, *RPE65*, *RDH5*, *RGR*, and *RLBP1* must be coordinated. As shown in Figure 6a, the proximal upstream region of these genes contains a binding site for the sex-determining Y region box (SOX).

The transcription factors SOX9 and LIM homeobox 2 (LXH2) are coexpressed in the nuclei of mature RPE cells [207]. As shown in Figure 6b, activating the RPE65 and RLBP1 promoters requires the synergic action of SOX9 and orthodenticle homeobox 2 (OTX2). In contrast, RGR promoter activation requires the synergic action of SOX9 and LXH2 [207].

Chromatin immunoprecipitation in human fetal RPE cells indicates that SOX9 and OTX2 bind the *RPE65*, *RGR*, and *RLBP1* promoter regions [207]. In mouse RPE cells, conditional *Sox9* inactivation causes a decrease in the expression of genes coding for retinoid cycle components, especially of *Rpe65* and *Rgr* [207]. Transcriptional control affecting the tissue-specific expression has been tracked to distinct elements within the 5′ untranslated region (UTR) of the *RPE65* coding sequence [208]. In HEK293 cells, an *RPE65*-expressing human cell line derived from the kidney, *RPE65* expression decreases in response to retinoic acid, which acts via several retinoid acid receptors (RAR) and retinoid X receptor (RXR) isoforms [209]. These results indicate the relevance of several transcription factors in coordinating the transcription of genes coding for visual cycle components. 

Although transcriptional control by several intrinsic and extrinsic factors allows the coordinated expression of genes coding for 11cis-RAL synthesis in RPE cells, RPE65 may represent the key target for controlling 11cis-RAL generation. As shown in Figure 6c, RPE65 operates at its V_max_ [13], indicating that 11cis-RAL synthesis may be controlled by the amount of translated protein rather than by substrate concentration. Indeed, there is evidence that posttranslational control over RPE65 adds up to the control at the transcriptional level. In a cell line model of human RPE (ARPE-19), the expression of several RPE genes involved in the 11cis-RAL synthesis, such as *LRAT*, *RPE65*, *RDH5*, *RGR*, *RDH10*, and *RLBP1*, increases significantly after four months compared to four days in culture [210], consistent with their coordinated transcriptional control during human RPE cell maturation. In addition to the transcriptional control, there is evidence for posttranslational mechanisms affecting the level of RPE65 protein, as ARPE19 cells lack RPE65 protein despite high-level *RPE65* expression. The finding of protein products coded by *LRAT*, *RDH5*, and *RLBP1* suggests a selective suppression of *RPE65* translation. Accordingly, incubation with at-ROL resulted in retinyl esters synthesis [210], indicating that the LRAT protein is functional. The failure to translate RPE65 is not due to ARPE19 cells, as no *RPE65* translation has also been reported for human iPS-derived RPE cells after several passages [211]. However, the control over RPE65 may depend on the specific cell lines used for hiPS cell generation, their culture conditions, and the time in culture. For instance, ARPE-19 cells cultured using a different protocol were reported to lack the expression of *LRAT*, *RPE65*, *RDH5*, and *RLBP1* [212], indicating the relevance of culture conditions for the proper expression of genes involved in the retinoid synthesis. The importance of culture time is indicated by the finding that human iPS-derived RPE cultured for up to 6 months expresses RPE65, translates it into a protein, and upon incubation with at-ROL, generates and releases 11cis-RAL into the culture medium [213]. 

The finding that transfection of human iPS-derived RPE cells with a lentivirus carrying RPE65 led to *RPE65* overexpression, but RPE65 increase was either not significant or barely biologically significant, depending on the assay used [214], which provides additional evidence for the control over RPE65 translation.

A translation inhibitory element has been reported within the *RPE65* 3′ UTR (Figure 6d), a 170 nucleotide sequence downstream of the stop codon [215]. The 3′ UTR is a target for microRNA (miRNA), and several miRNAs have been reported to affect RPE cell differentiation and survival (reviewed in [216,217]). Among several miRNA-modulating RPE cells’ gene expression, miRNA-410 has been reported to target *RPE65* and *OTX2*, whose expression increases in response to miRNA-410 inhibition [218]. 

Additional factors modulating RPE65 activity at the posttranslational level may include its palmitoylation [219] and interactions with inhibitory proteins, such as FATP1 [220,221] (reviewed in [126]). The functional role of these interacting proteins is presently unclear, as they would not affect the amount of available RPE65, as reported in cultured human RPE cells. However, FATP1 has been reported to affect RPE65 enzymatic activity and photoreceptors’ OS length [221]. The extent of control over RPE65 translation suggests cells limit 11cis-ROL synthesis to avoid exceeding 11cis-RAL levels. As noted by [214], this control may represent a safety mechanism for patients treated with viral vectors carrying *RPE65* for the first approved gene therapy of biallelic pathogenic variants of RPE65 [222,223,224]. It is unclear whether the translational control of RPE65 reported for lentivirus-transfected *RPE65* [214] will also operate for the adenovirus-transfected *RPE65* used for gene therapy [222,223,224].

### 4.3. Retinoids and Photoreceptors Maturation and Viability

Besides their retinal-specific control of light responsiveness, retinoids may affect retinal development and viability [225]. In addition to 11cis-RAL, the RPE may also generate other RAL isomers. A technical issue should be mentioned in addressing this point, i.e., that photochemical isomerization of at-RAL may generate 9cis-, 11cis-, and 13cis-RAL (see Figure 9 in [11]). Consequently, it could be difficult to unambiguously attribute low 9cis-RAL levels to endogenous synthesis by an enzymatic pathway rather than photoisomerization of at-RAL. However, there is evidence that RPE cells may generate low levels of 9cis-RAL in *RPE65*^−/−^ mice. When these mice were raised in cyclic light, they did not generate 11cis-RAL, and rod photoreceptors lacking rhodopsin did not respond to light stimuli. However, when raised for four weeks in darkness, their RPE cells may generate 9cis-RAL, and rod photoreceptors respond to light stimuli using isorhodopsin, representing 9cis-RAL-bound opsin [226]. Considering that 9cis-RAL production is only found in these mice when raised in darkness, the photoisomerization of at-RAL does not seem to be a likely explanation. Additional evidence that 9cis-RAL synthesis occurs via an enzymatic pathway stems from the finding that the amount of 9cis-RAL depends on mice’s genetic background [227]. These results are consistent with RPE cells synthesizing and releasing 9cis-RAL for transport along the SRS to photoreceptors for opsin regeneration. The finding that 9cis-RAL is not observed when RPE65 wt mice are raised for four weeks in darkness [226] may indicate that RPE65 suppresses 9cis-RAL synthesis. It is, however, unclear whether RPE cells generate 9cis-RAL or pick it up from plasma. In culture media of human iPS-derived RPE cells incubated with 10 µM at-ROL, a peak corresponding to authentic 9cis-RAL suggests that these cells may generate 9cis-RAL from at-ROL [213]. However, the authors suggest that the peak may result from the photoisomerization of 11cis-RAL [144], considering that 9cis-RAL is not believed to have physiological significance in the eye. 

However, evidence is consistent with a role for 9cis-ROL derivatives in the retina/RPE. The synthesis of 9cis-RAL has been shown to occur from 9cis-ROL via RDH4, a retinol dehydrogenase with selectivity for 9cis-ROL over at-ROL [228]. *RDH4* expression in the mouse retina and RPE is already present on gestational day 11 [228], well before *RDH5*, whose expression starts in the postnatal retina (see Figure 6 in [229]). In adult mice, *RDH4* expression shows the highest expression in the RPE and ONL [229]. The biological relevance of RDH4 during development may be linked to 9cis-RA generation, the agonist of the retinoid X receptors (RXR) selectively activated by 9cis-RA over at-RA (reviewed in [230]). 

Data indicating that RPE cells had already developed pigmentation by FW 7–8 in humans suggests that their differentiation starts at least ten weeks before rods start expressing opsin. However, data indicating that by 26 weeks in culture human iPS-derived RPE cells generate and release 11cis-RAL in the medium [213] suggest a synchronization between 11cis-RAL synthesis by RPE cells and opsin incorporation into OS. This synchronization may match 11cis-RAL in RPE cells with OS generation by photoreceptors, preventing the accumulation of unliganded opsin suggested to affect rod viability adversely [231]. Moreover, recent findings in adult mice showing that reducing 11cis-RAL synthesis may improve cone viability following rod loss [58] may indicate the importance of preventing 11cis-RAL increase during development, when cones’ birth precedes that of rod photoreceptors. It is presently unclear how RPE and photoreceptors attain their synchronization. Section 4.1 discussed the evidence for posttranslational mechanisms affecting RPE65 levels, indicating that RAR and RXR activation inhibit RPE65 expression by HEK293 cells [209]. In this case, 9cis-RA derived from 9cis-RAL may reduce *RPE65* expression, but the functional relevance compared to translational control over *RPE65* messenger is unclear, considering that translation control of RPE65 may represent the primary regulatory mechanism, as discussed in Section 4.1. A different mechanism involves RAR and RXR control on the expression of *NRL*, which represents a major transcriptional factor driving rhodopsin expression [232]. RA and 9cis-RA activate *NRL* expression in human Y79 neuroblastoma cells, and similar actions were observed in cultured rat and porcine photoreceptors [233]. RA-responsive elements (RAREs) were identified in the *Nrl* promoter region of the bovine retina, suggesting that 9cis-RA may affect rod photoreceptor maturation [233]. In addition, 9cis-RAL has been reported to accelerate the development of human iPS cell-derived retinal organoids (RO) compared to at-RA [234,235].

Interestingly, human iPS-derived RO exposed to RA from day 65 to day 120 has been reported to show delayed development compared to those cultured without RA [236]. However, 1 µM RA-treated human iPS cell-derived RO developed thicker ONL with increased rod photoreceptors than those cultured without RA [236], making it unclear whether the accelerated development observed with 9cis-RAL instead of RA provides an advantage for rod photoreceptors maturation. Considering that, compared to low (0.5 µM) RA [237], high (10 µM) RA decreased RPE pigmentation and inhibited the maturation of photoreceptors, 9cis-RAL may improve photoreceptor differentiation by having a weaker affinity for RXR receptors than 9cis-RA. 

The evidence indicates that 9cis-ROL derivatives may build up in the RPE of dark-adapted mice lacking RPE65, suggesting specific roles in the eye, such as controlling opsin expression in maturing and adult rod photoreceptors. It is possible that 9cis derivatives, produced by RPE cells, could impact rod maturation and opsin levels, potentially coordinating 11cis-RAL synthesis by RPE with opsin expression in rods. 

However, there is a lack of direct experimental evidence supporting this hypothesis. For example, it is uncertain whether selectively blocking RDH4 expression in the RPE would affect rod photoreceptors’ maturation and opsin expression or the activation of genes controlled by the Nrl promoter. Another point needing clarification is whether the synthesis of 9cis-ROL derivatives is suppressed by 11cis-RAL synthesis, which could balance 11cis-RAL synthesis and opsin expression by rods.

## 5. Conclusions

Over the past two decades, several significant advancements have reshaped our understanding of 11cis-RAL synthesis. One of the key breakthroughs was the discovery of RPE65 as the enzyme responsible for the conversion of at-ROL to 11cis-ROL by RPE cells [13,14]. Additionally, the role of RGR in the photoconversion of at-RAL to 11cis-RAL by RPE and MGCs [52,53,54] has also been recognized as a significant advancement. 

Other notable progress stemmed from investigations into the impact on vision and retinal cell viability of gene variants impairing the operation of specific components of retinoid biosynthetic pathways. Although defects in the conversion of at-RAL released from bleached opsin to 11cis-ROL have been associated with the generation of toxic byproducts that eventually accumulate in RPE cells [79,103], it is unclear why this would lead to early retinal dystrophies affecting cone cells. However, as discussed in Section 3, recent evidence suggests that the loss of RPE pigmentation precedes an increase in bisretinoids fluorescence [160,161], suggesting that in vivo bisretinoids may associate with retinal dystrophies but not necessarily have a causal role in the early loss of RPE and photoreceptor cells via their oxidation by blue light.

Indeed, gene variants unrelated to the retinoid cycle that affect lipid handling in RPE cells are also associated with retinal dystrophies involving early cone loss [134,138,142,144,150,157,236] and the metabolic reprogramming of RPE cells [144,157]. This evidence indicates that A2E may impact RPE cells by impairing lipid handling [139,140,141] and activating innate immunity [144,145,147,148,149]. Considering that pigmentation is an early sign of RPE differentiation, bisretinoid accumulation in photoreceptors may follow a metabolic dysregulation of lipid handling in RPE that impairs glucose supply to photoreceptors, reducing their ability to reduce at-RAL to at-ROL via NADP-dependent enzymes [32,65]. The recent evidence that peripheral cones survive in the retina after rods have degenerated indicates that the metabolic support provided to cones by rods is likely to be critical when rods and cones compete for glucose provided by the RPE. However, cones may survive after rod degeneration by MGCs activating the signaling pathway associated with RA synthesis [202]. 

At the same time, pathogenic variants reducing 11cis-RAL also cause early-onset retinal dystrophies, which are currently explained as a result of the persistent activation of the phototransduction cascade by unliganded opsin. The recent finding that RGR-dependent photoconversion of at-RAL to 11cis-RAL in RPE cells also contributes to the ligand for rhodopsin regeneration in darkness [54,88,89] may indicate that some opsin may recover to the ground state when 11cis-RAL synthesis via RPE65 is blocked. In addition, in darkness, 9cis-RAL synthesis may provide the ligand for isorhodopsin synthesis [226]. It is presently unclear whether a threshold level of unliganded opsin triggers photoreceptor loss and, in this case, its value. Recent findings showing improved cone survival following the suppression of 11cis-RAL synthesis [58] after the rod had degenerated may indicate that the critical level of 11cis-RAL synthesis may be variable, depending on the amount of unliganded opsin available to bind 11cis-RAL.

The notion of a potential connection between 11cis-RAL synthesis by RPE cells and opsin levels in photoreceptors is intriguing but currently lacks experimental evidence. In Section 4.2, we reviewed the evidence suggesting that the regulation over 11cis-RAL synthesis in RPE cells may occur at the posttranscriptional level, potentially through a control on RPE65 transcript translation. In rods, regulation of opsin levels may occur at the transcriptional level via the transcription factor NRL, whose expression is promoted by RA [233]. In cones, the control over opsin levels may also occur at the transcriptional level via thyroid hormone receptors, which may heterodimerize with RXR receptors [238]. Interestingly, RXR receptors respond to 9cis-RA, which promotes NRL expression in rods [233], suggesting that 9cis-RA may direct opsin expression in rods and cones. In this regard, 9cis-RAL has been found to promote the differentiation of human iPS cell-derived RO more effectively than RA [235]. Considering the ability of 9cis-RAL to regenerate opsin and promote RO differentiation more effectively than RA, it is unclear whether the protection afforded by 9cis-RAL treatment or its pro-drug against retinal dystrophies [129,130,131] is entirely due to its ability to remove unliganded opsin, or instead reflect its action at the transcriptional level. The observation that isorhodopsin formation could only be observed in animals lacking RPE65 activity after four weeks of dark adaptation, and the amount of isorhodopsin generated is relatively low, suggesting that 9cis-RAL synthesis may occur at a low rate. It is unclear whether this low rate of 9cis-RAL synthesis in the RPE via RDH4 may have functional relevance. However, it should be considered that 9cis-RAL may have a higher affinity for RXR than for opsins, and opsin binding sites vastly exceed those of RXR, suggesting that it may have a functional impact on transcription at a concentration that would only bind a tiny fraction of unliganded opsins.

In conclusion, recent evidence indicates that it is crucial to regulate 11cis-RAL synthesis to match the amount of opsin in photoreceptors. However, our understanding of the mechanisms that govern 11cis-RAL synthesis and its coordination with opsin expression is still limited. Future research exploring how these processes are coordinated could enhance our knowledge of how outer retinal cells operate in visual transduction. Improved knowledge, in turn, may lead to the development of innovative approaches to counteract the loss of photoreceptors in retinal dystrophies of genetic origin.

## Figures and Tables

**Figure 1 cells-13-00871-f001:**
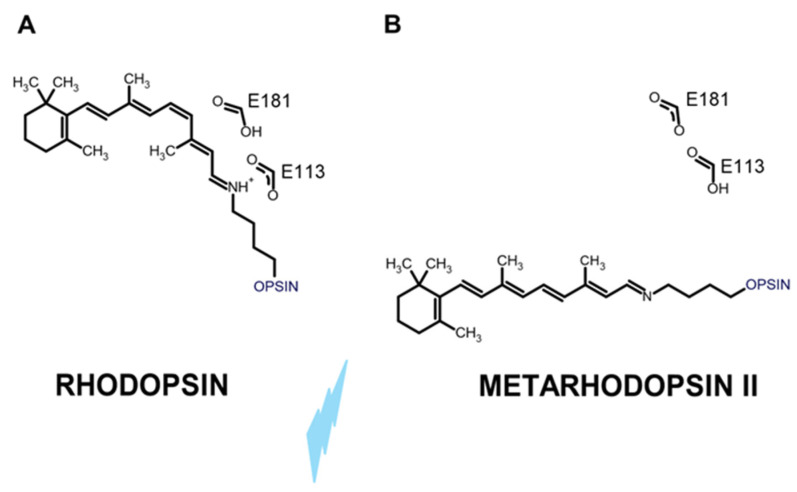
(**A**) In ciliary photoreceptors expressing c-type opsin with bound 11cis-RAL, the negative charge provided by E113 stabilizes the positive charge of the protonated Schiff base. (**B**) Light (cyan) isomerizes 11cis-RAL to at-RAL. After several quick rearrangements, E133 may not stabilize the metarhodopsin II deprotonated Schiff base, which became protonated, preventing at-RAL photoreversal to 11cis isomers.

**Figure 2 cells-13-00871-f002:**
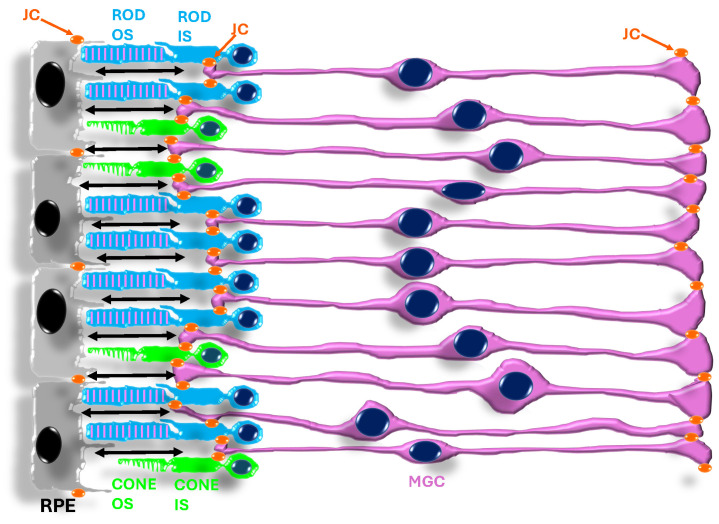
Outer (OS) and inner (IS) segments of cone (green) and rod (cyan) cells lie in the subretinal space (double-arrow line), which is limited by the junctional complexes (orange) of the retinal pigment epithelium (RPE) and those of the outer limiting membrane (OLM) between Müller glial cells (MGCs) (pink) and rod and cone IS.

**Figure 3 cells-13-00871-f003:**
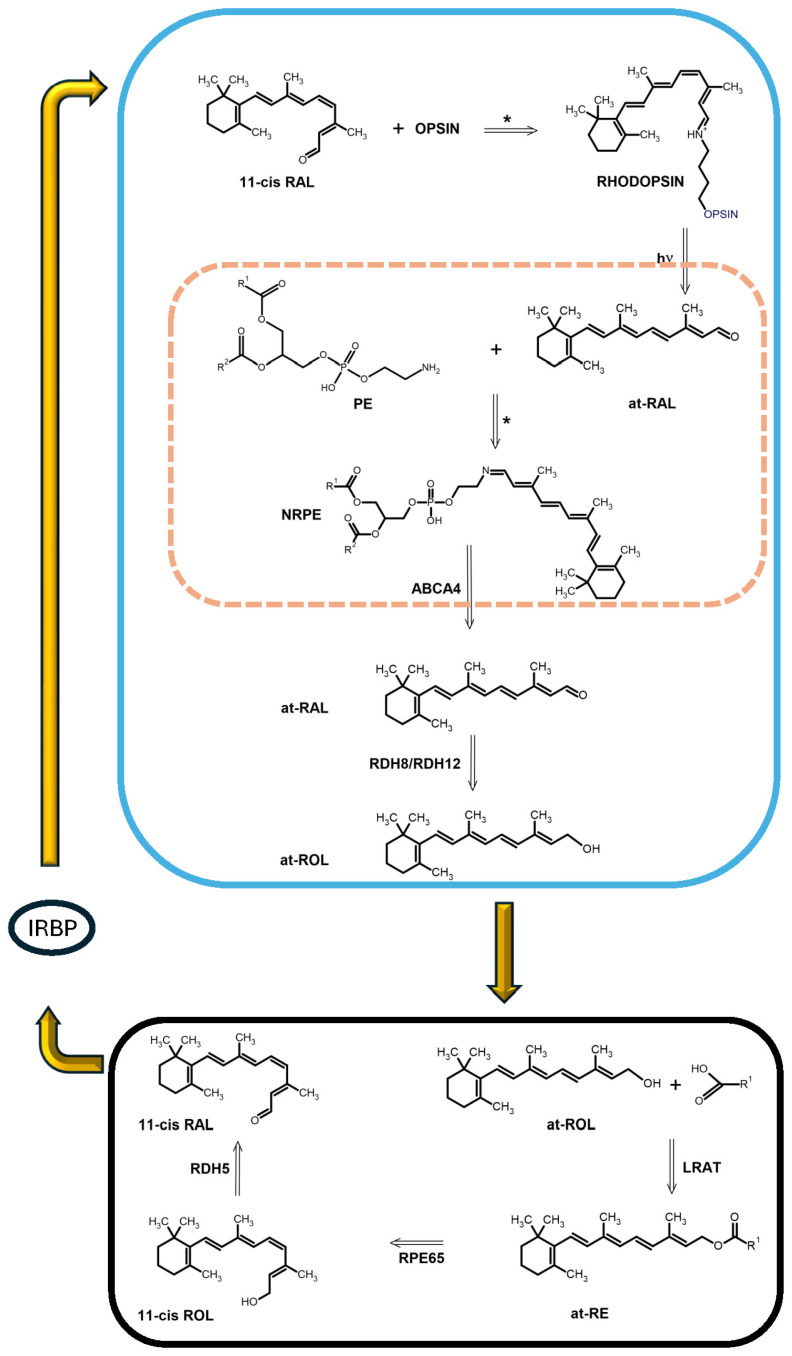
The cyan box shows chemical events in the OS. Opsin and 11cis-RAL react, forming a protonated Schiff base between a lysin amino group of the opsin and the aldehyde. After light-induced (hv) 11cis-RAL isomerization, at-RAL is released inside the disk (orange dotted box). It reacts spontaneously with the phosphatidyl ethanolamine (PE) amino group, generating N-retinylidene ethanolamine (NRPE). NRPE is transferred to the OS cytoplasm by the flippase ABCA4. at-RAL dissociates from PE in the cytoplasm and is reduced to all-trans retinol (at-ROL or vitamin A) by retinol dehydrogenase 8 and 12 (RDH8 and RDH12). In the RPE (black box), retinol generated and released by the OS, is esterified by lecithin retinol acyl transferase (LRAT) into at-RE. RPE65 exhibits isomerohydrolase activity, converting at-RE into 11cis-ROL. Retinol dehydrogenase 5 (RDH5) converts 11cis-ROL into 11cis-RAL, which may be bound by cellular retinaldehyde-binding protein (CRALBP) (not displayed in the Figure) to prevent spontaneous isomerization. CRALBP eventually transfers 11cis-RAL to the interphotoreceptor-binding protein (IRBP), which conveys it to the OS by diffusion across the subretinal space (yellow arrows). * indicates a spontaneous reaction between amino groups and an aldehyde group.

**Figure 4 cells-13-00871-f004:**
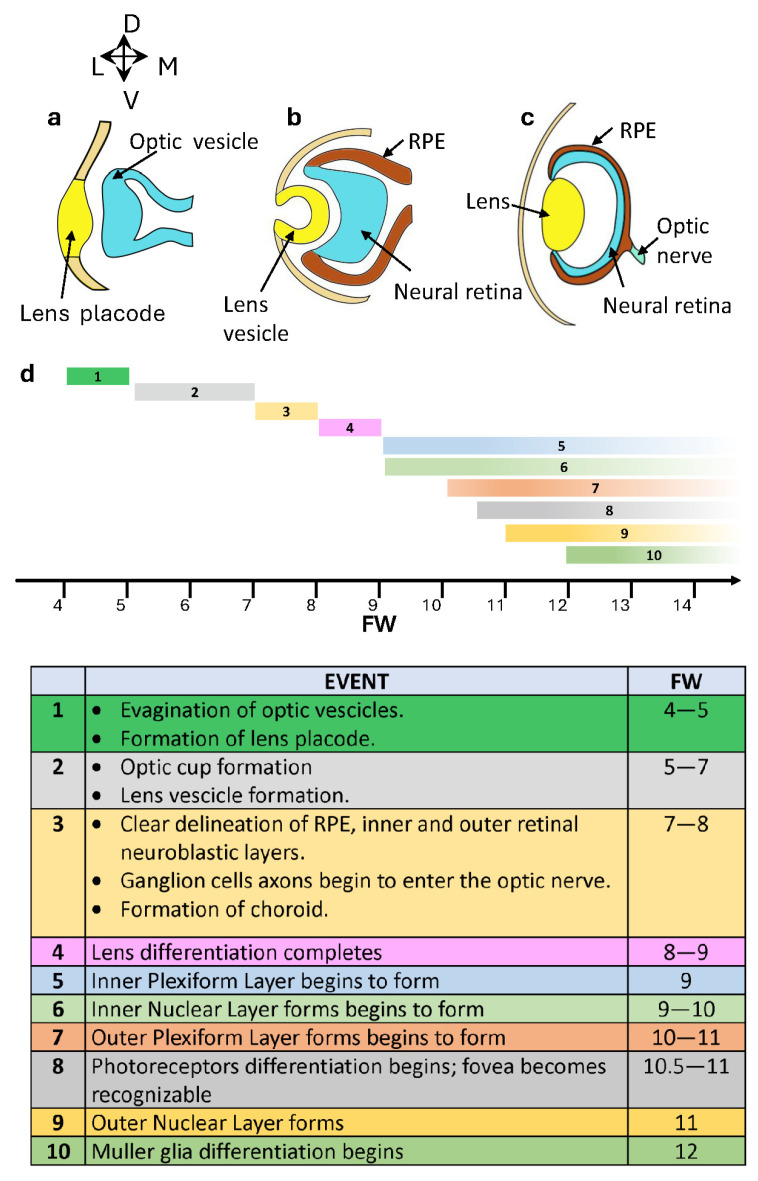
Schematic representation of critical phases in human eye development. (**a**) Formation of the optic vesicle and the lens placode. (**b**) Progression in optic cup and lens vesicle development, with simultaneous specification of the neural retina and the RPE. (**c**) The advancements in organized eye structure involve neural retina stratification and optic nerve growth. (**d**) Timeline and table of the 10 main developmental steps of the human retina from FW4 to FW12. As the table below indicates, steps 1–4 occur within the timeline. Steps 5–10 start at the indicated time at the location of the presumptive foveal. They will spread in the same order to adjacent retinal areas, reaching the peripheral retina around FW 40.

**Figure 5 cells-13-00871-f005:**
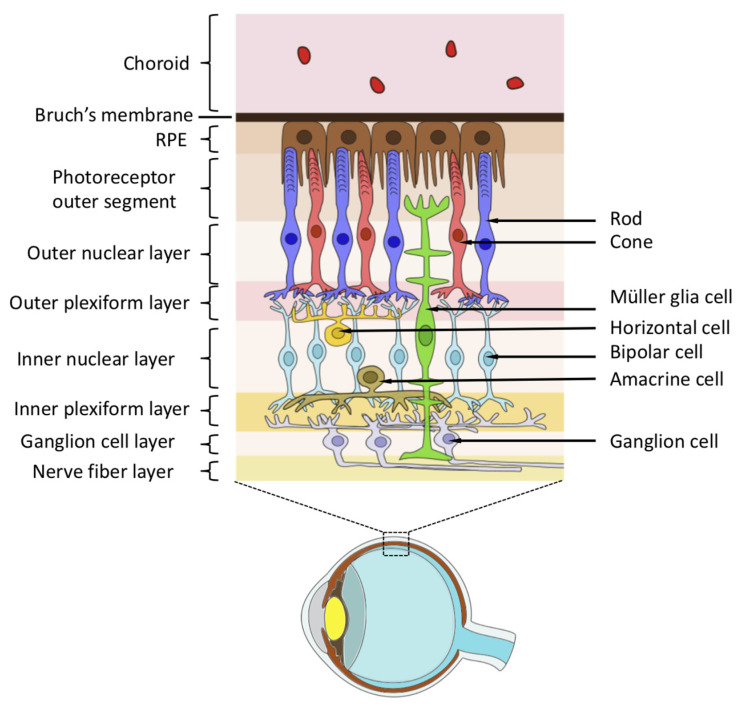
Structure of the mature retina.

**Figure 6 cells-13-00871-f006:**
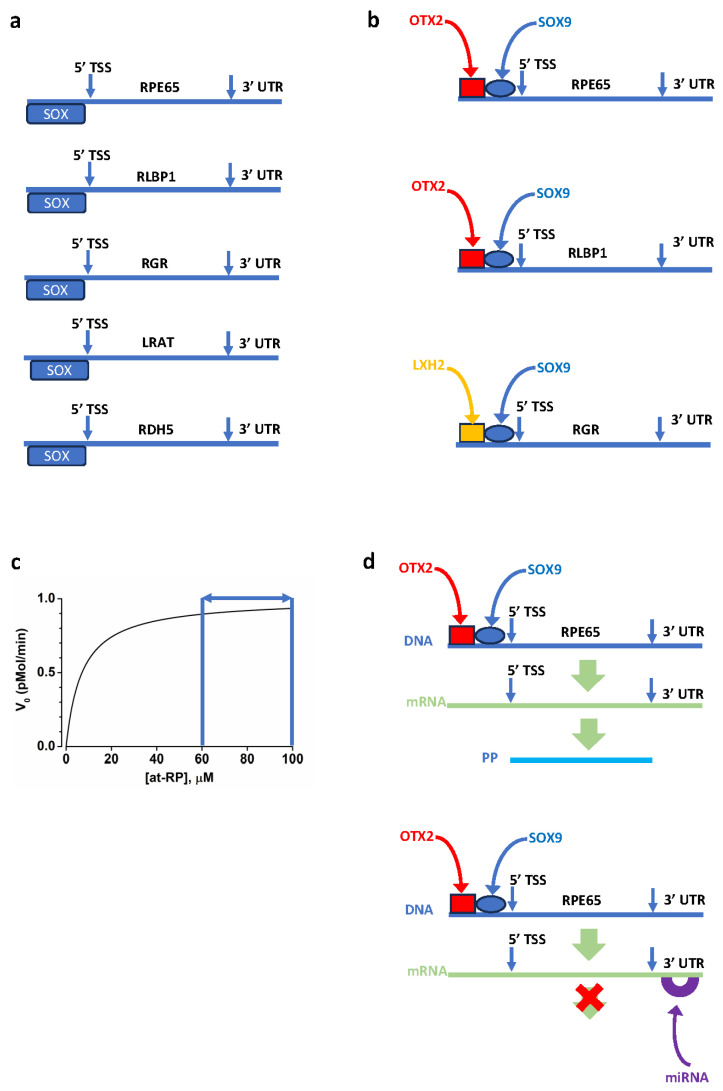
(**a**) A SOX binding site in the proximal region upstream of the 5′ transcription start site (TSS) of *RPE65*, *RLBP1*, *RGR*, *LRAT*, and *RDH5*. (**b**) Transcription factors SOX9 (blue oval) and OTX2 (red square) bind the promoter region of *RPE65* and *RLB1*; SOX9 and LXH2 bind the *RGR* promoter region. (**c**) The black curve plots RPE65 isomerohydrolase activity as a function of all-trans-retinol palmitate (at-RP) concentration. The horizontal blue double-arrow segment indicates the substrate concentration range in the RPE, indicating the enzyme operates at its maximal velocity. (**d**) Translational control involves microRNA (miRNA) (purple half circle) binding at the 3′ untranslated region (3′ UTR) of *RPE65* mRNA to prevent its translation (red cross superimposed on the green arrow) into PP.

**Table 1 cells-13-00871-t001:** First column: human gene names. Second column: cells expressing genes listed in the first column. The third, fourth, and fifth columns report the total, the pathogenic, and likely-pathogenic variants for each gene, respectively. Last column: length (in AA) of the human proteins.

Gene Name	Expressing Cells	Total Variants	Pathogenic	Likely Pathogenic	AA Residues
*ABCA4*	Rods, Cones, RPE	3252	787	480	2273
*RDH8*	Rods, Cones	9	0	0	311
*RDH12*	Rods, Cones	488	83	66	316
*LRAT*	RPE	242	39	16	230
*RPE65*	RPE	800	175	87	533
*RDH5*	RPE	264	43	6	318
*RLBP1*	RPE, MGCs	271	13	13	317

**Table 2 cells-13-00871-t002:** Transgenic mouse models to address genes’ function in wt cells and phenotype in KO mice.

Mouse	Expressing Cells	Gene Function	Phenotype
*Rgr^−/−^* KO	RPE, MGCs	Retinal G-protein receptor—light-induced at-RAL conversion into 11cis-RAL.	Reduced light responsiveness in light-adapted mice.
*Nrl^−/−^* KO	Rods	Leucine zipper transcription factor—promotes rod gene expression and suppresses cone genes.	Rod-fated cells keep expressing sw opsin and cone-specific genes, resulting in an excess of blue-cone-like photoreceptors that degenerate.
*Gnat1^−/−^* KO	Rods	Guanine nucleotide alpha transducing—increases cGMP phosphodiesterase activity in response to rhodopsin activation.	Mice lack rod photoresponse, allowing the analysis of cone light response. No effect on rod viability.
*Alms1^−/−^* KO	Ciliated cells	Alstrom syndrome 1 may affect cilia function.	Rod-cone degeneration, hearing loss, and multi-organ syndrome.
*Abca4^−/−^* KO	Rods, cones, RPE cells	ATP-binding cassette subfamily A member 4—flippase translocates at-RAL to promote its detoxification by its reduction to at-ROL.	Increased accumulation of fluorescent derivatives of at-RAL. Slow photoreceptor degeneration in mice.

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
