# Peer review of "Retinoid Synthesis Regulation by Retinal Cells in Health and Disease"

_cells, 2024, doi:10.3390/cells13100871_

Round 1
Reviewer 1 Report
Comments and Suggestions for Authors
I was delighted to read this paper. Absolutely fine. There is one point I would add to the current version: how electrical activity chages with retinois synthesis/content/metabolism. I find it inportant to the readers of physiological interest that this issue is addressed because overall retinal function is manifested in converting light to electric impulses. Changes in the state of light absorbing molecules szch as retinoids are of primary importance in this process.
Author Response
COMMENT: There is one point I would add to the current version: how electrical activity chages with retinois synthesis/content/metabolism. I find it inportant to the readers of physiological interest that this issue is addressed because overall retinal function is manifested in converting light to electric impulses. Changes in the state of light absorbing molecules szch as retinoids are of primary importance in this process.
A: We have added a sentence (l. 42-44) in the revised version to remind readers that light absorption leads to an electrical response in both r- and c-type photoreceptors.
Moreover, we have specified the impact of dysregulated retinoid synthesis on the recovery of electrical responsiveness of vertebrate photoreceptors in several instances, as detailed:
- 264-271. Describes Rgr's role in supporting the electrical response of mouse cone photoreceptors during continuous exposure to bright light
- 299-303. These sentences describe the faster recovery of cone light response in the isolated retina of mice lacking the Alms protein. These mice reproduce the early cone-rod dystrophy of ALMS1 patients. Data suggest the defect lies in the unregulated synthesis of 11cis-RAL via Rgr-mediated at-RAL photoconversion.
- 416-428. In this paragraph, we discussed recent evidence addressing the notion of rod saturation in bright light. Rod saturation means that all light-sensitive channels have been closed in response to a light stimulus, and an increase in light intensity could not elicit any electrical response. However, rod saturation does not indicate that all opsins have been bleached. We report data showing that the rod photoreceptors recover their ability to generate an electrical response to light after 10-20 minutes of continuous bright light exposure “in vivo” and “in vitro” due to adaptation mechanisms involving transducin translocation from the OS to the IS.
Reviewer 2 Report
Comments and Suggestions for Authors
This review of opsin chromophore metabolism in terrestrial vertebrates is a wide-ranging compilation that will be of broad interest, particularly for vision scientists.
Only a few comments/concerns:
Interphotoreceptor retinoid binding protein is generally abbreviated IRBP, not IPRBP or IPRB as currently used in the manuscript.
There are several paragraphs, or large regions of some paragraphs, that do not include appropriate citations to the original studies leading to the relevant statements. As currently written, it would be very difficult for a reader to figure out how to access the original studies. Examples of parts of the manuscript that are missing citations: lines 181-190; lines 215-218; lines 402-407; lines 489-494 (generic reviews cited only); lines 501-517; lines 573-584; and lines 588-592.
Author Response
This review of opsin chromophore metabolism in terrestrial vertebrates is a wide-ranging compilation that will be of broad interest, particularly for vision scientists.
Only a few comments/concerns:
Q1: Interphotoreceptor retinoid binding protein is generally abbreviated IRBP, not IPRBP or IPRB as currently used in the manuscript.
A1: In the revised version, the typos have been corrected to the standard form IRBP
Q2: There are several paragraphs, or large regions of some paragraphs, that do not include appropriate citations to the original studies leading to the relevant statements. As currently written, it would be very difficult for a reader to figure out how to access the original studies. Examples of parts of the manuscript that are missing citations: lines 181-190; lines 215-218; lines 402-407; lines 489-494 (generic reviews cited only); lines 501-517; lines 573-584; and lines 588-592.
A: In the revised version, we have added new citations as detailed:
lines 186-194; References 14, 31-36 have been added (l. 186-195 of the revised version)
lines 215-218; References 20, 30, 37 have been added (l. 221-226 of the revised version)
lines 402-407; References 37, 168-170 have been added (l. 680-686 of the revised version)
lines 489-494 (generic reviews cited only); References 189-191 have been added to references 192-193 (l. 771-786 of the revised version).
lines 501-517; References 194-196 have been added (l. 787-798 of the revised version)
lines 573-584; References 75, 200-203 have been added (l. 857-869 of the revised version)
lines 588-592; Reference 205 has been added (l. 872)
Reviewer 3 Report
Comments and Suggestions for Authors
Comments and Suggestions for Authors
The perspective discusses the regulation of 11-cis-retinal concerning photoreceptor development and survival. This shift focus from the current perspective that bis-retinoid products generated from excess all-trans-retinal following visual transduction drive the macula degeneration observed in retinal diseases such as Stargardts. However, there are some further considerations which need to be considered to convincingly challenge the evidence of bisretinoid accumulation from vitamin A derivatives driving RPE degeneration at the macula and subsequently photoreceptor cell loss. Specific comments for the authors are as follows.
Page 5 Figure 2 legend. Several uses of “x convert in y” this should be amended to “x converted to y” or “x converted into y”.
Page 5 line 201. “Several tenths minute” is unclear is this “several tenths of a minute”.
Page 6 line 226. “despite RGR lacks…” should be “despite RGR lacking…”.
Page 6 line 239-241. The sentence needs rephrasing. Also, does RGR directly interact with RPE65 or indirectly stimulate isomerohydrolase activity of RPE65?
Page 6 line 244. RDH has not previously been used to define abbreviation.
Page 7 line 277. “to11cis RAL” “to 11cis-RAL”.
Page 7 line 291. “photoconversion of at-RAL in 11cis-RAL” should be “photoconversion of at-RAL to 11cis-RAL”.
Table 1. Second row for ABCA4. ABCA4 is also expressed in the RPE. DOI: 10.1073/pnas.1802519115 and DOI: 10.1016/j.exer.2020.108204.
Page 8 line 333-334. The reaction of NRPE with at-RAL produces A2PE and subsequently hydrolysed to A2E accumulating in the RPE.
Page 8 line 340-343. The statement before [64] is unrelated to after. The low A2E in cells with low synthesis of 11-cis-retinal is most likely due to the reduced visual transduction and release of all-trans-retinal precursor to A2PE>>A2E. Similarly, how bright light can increase A2E as this triggers more visual transduction and release of all-trans-retinal.
Page 8 line 345-348. The lower proportion of PE in cones could increase their vulnerability to forming condensed products of all-trans-retinal and NRPE (bisretinoid pre-cursors) doi: 10.1007/s00417-022-05684-9. This is because of an excess of all-trans-retinal to NRPE, cones express more ABCA4. But pathogenic ABCA4 variants delay clearance of NRPE which allows the all-trans-retinal in excess to react with the lingering NRPE. The increased vulnerability of cones may also be caused by their concentration at the macula where the structural adaption in the macula causes a higher demand on the RPE which supports them.
Page 8 line 356-358. This does not prove that A2E accumulation is not driving pathology, this evidence in [69] shows mechanistically how A2E can cause photoreceptor loss, and how this may be alleviated by vitamin E, protecting against A2E-oxidation.
Page 8 line 376. This needs to be expanded to counter current work suggesting dysregulation of the retinoid cycle drives pathology in the RPE by Radu’s group. doi: 10.1073/pnas.1802519115. There is also the consideration that without a macula, as seen in abca4 -/- mouse models lipofuscin is detected but not thinning of the retina as the photoreceptor and RPE does not atrophy.
Page 9 line 416. This section uses “11-cis RAL” where previously uses “11 cis-RAL”.
Page 10. Figure 3. Panel d is identified as c.
Page 11 line 445-447 11-cis RAL and 11 cis-RAL are both being used.
Page 11 line 451 “1-cis RAL”
Page 17 line 724 “that translational control on RPE65” should be “that translational control of RPE65”.
Page 17 line 761-762 “11-cis RAL” and “11-cis-ROL”. Should these be “at-RAL” and “at-ROL”.
Page 17 line 761-764 (some further considerations mentioned as per page 8) Whilst RPE atrophy does proceed photoreceptor loss in stargardt’s disease there is a clear implication of retinoid dysregulation driving the disease since it is caused by reduced function of ABCA4. Current work does show ABCA4 may have additional roles, including in the RPE, but these still appear to be related to excessive all-trans-retinal or NRPE. There would need to be a proposed link in the RPE dysregulation / reduced metabolism caused which makes sense in the context of this genetic background, which has previously provided support to the bisretinoid driving the RPE loss and subsequent photoreceptor loss. There may be a higher metabolic demand on the macula where any dysfunction in the RPE compromises the ability to maintain itself and the photoreceptors it supports.
Comments on the Quality of English Language
Some minor English language amendments have been highlighted in the comments and suggestions for Authors with page and line to help locate.
Inconsistencies with abbreviations need to be standardised throughout the perspective for the final draft (if this work is to be considered again after major revisions).
Author Response
The perspective discusses the regulation of 11-cis-retinal concerning photoreceptor development and survival. This shift focus from the current perspective that bis-retinoid products generated from excess all-trans-retinal following visual transduction drive the macula degeneration observed in retinal diseases such as Stargardt. However, there are some further considerations which need to be considered to convincingly challenge the evidence of bisretinoid accumulation from vitamin A derivatives driving RPE degeneration at the macula and subsequently photoreceptor cell loss. Specific comments for the authors are as follows.
We thank the reviewer for the time and effort she/he put in indicating several typos and mistakes in the original version. We also appreciated the constructive criticisms raised. Before addressing the specific queries, we would like to point out that the main issue we wished to address was not whether bisretinoids play a role in macular dystrophies but how they affect macular photoreceptors.
Q1: Page 5 Figure 2 legend. Several uses of “x convert in y” this should be amended to “x converted to y” or “x converted into y”.
A1: In the revised version, these errors have been amended in Figure 2 and through the text.
Q2: Page 5 line 201. “Several tenths minute” is unclear is this “several tenths of a minute”.
A2: In the revised version, the sentence now reads …. several tenths of a minute….. (l. 207).
Q3: Page 6 line 226. “despite RGR lacks…” should be “despite RGR lacking…”.
A3: In the revised version, the sentence now reads …. despite RGR lacking ……. (l. 232).
Q4: Page 6 line 239-241. The sentence needs rephrasing. Also, does RGR directly interact with RPE65 or indirectly stimulate isomerohydrolase activity of RPE65?
A4: In the revised version, the sentence reads, “However, Rgr-/- mice had less than half the isomerohydrolase activity of wt mice, suggesting that RGR may stabilize RPE65 isomerohydrolase activity [47]. RGR has been reported to coprecipitate with RDH5 and RPE65 [48], the isomerohydrolase of the retinoid cycle [13,14], although evidence for a direct interaction has only been found for RDH5 and RPE65 [48]. ” (l. 243-247).
Q5: Page 6 line 244. RDH has not previously been used to define abbreviation.
A5: In the revised version, p. 5, lines 193-194, describe the conversion of at-RAL to 11-cis-ROL by retinol dehydrogenase 5 (RDH5).
Q6: Page 7 line 277. “to11cis RAL” “to 11cis-RAL”.
A6: In the revised version, the typo has been corrected (l. 283).
Q7: Page 7 line 291. “photoconversion of at-RAL in 11cis-RAL” should be “photoconversion of at-RAL to 11cis-RAL”.
A7: In the revised version, the error has been corrected (l. 281).
Q8: Table 1. Second row for ABCA4. ABCA4 is also expressed in the RPE. DOI: 10.1073/pnas.1802519115 and DOI: 10.1016/j.exer.2020.108204.
A8: TABLE 1 reports RPE in the second row in the revised version.
Q9: Page 8 line 333-334. The reaction of NRPE with at-RAL produces A2PE and subsequently hydrolysed to A2E accumulating in the RPE.
A9: In the revised version, the sentence has been modified and now reads, “As individuals age, lipofuscin deposits begin to accumulate in the RPE [74]. Lipofuscin deposits contain bisretinoids, such as retinylidene-N-retinyl ethanolamine (A2E)[75,76]. A2E results from the reaction of a second at-RAL with NRPE to generate A2PE [75,76], which is hydrolyzed to A2E in the RPE [75,76]. ” (l. 354-358)
Q10: Page 8 line 340-343. The statement before [64] is unrelated to after. The low A2E in cells with low synthesis of 11-cis-retinal is most likely due to the reduced visual transduction and release of all-trans-retinal precursor to A2PE>>A2E. Similarly, how bright light can increase A2E as this triggers more visual transduction and release of all-trans-retinal.
A10: In the revised version, the sentence has been modified and reads, “The finding of decreased A2E derivatives in mice with reduced synthesis of 11cis-RAL by RPE cells [78] is most likely due to the reduced at-RAL release from opsins and the decreased A2PE generation that will suppress A2E build-up in RPE cells. Evidence suggests that at-RAL condensation products (bisretinoids) may act as photosensitizers that trigger oxidative stress in RPE and retinal cells in response to light (recently reviewed in [79]). ” (l. 365-371).
Q11: Page 8 line 345-348. The lower proportion of PE in cones could increase their vulnerability to forming condensed products of all-trans-retinal and NRPE (bisretinoid pre-cursors) doi: 10.1007/s00417-022-05684-9. This is because of an excess of all-trans-retinal to NRPE, cones express more ABCA4. But pathogenic ABCA4 variants delay clearance of NRPE which allows the all-trans-retinal in excess to react with the lingering NRPE. The increased vulnerability of cones may also be caused by their concentration at the macula where the structural adaption in the macula causes a higher demand on the RPE which supports them.
A11: We acknowledge that the reviewer’s point makes sense. However, evidence in reconstituted membranes (no ABCA4) suggests that PE rather than at-RAL is the critical factor, as the amount of A2PE does not increase when the at-RAL to PE ratio is above 2. Instead, DHA-containing PE generates less A2PE than DHA-lacking PE, suggesting that their higher DHA content protects rods despite their higher PE rods, and cones have to increase ABCA4 to compensate for their low DHA content. In the revised version, we discussed this point along with the data from the pure cone retina of double mutant mice lacking Abca4 and Nrl, which is often quoted as evidence that cones have a higher A2PE synthesis than rods. However, the raw data showed higher A2E content in rod-dominant than cone-dominant retinas lacking ABCA4. However, the normalization to 11cis-RAL content, aimed at compensating for the lower opsin content of the cone-dominant retinas, generates a 6-fold increase in the A2PE content of the cone-dominant retina. The normalization assumes that rod- and cone-dominant retinas have the same number of OS, which is not true (the cone-dominant retina has 60% of the OS of the rod-dominant retina, see ref 84 Daniele et al. 2005 and ref 87 Nikonov et al., 2006).
Moreover, the cone OS volume is about ¼ of the rod OS (see Table 1 in ref 87). After correcting for these factors (1/(0.6x0.25)=6.7, the amount of 11cis-RAL for OS is similar in rods and cones, making the scaling of cone A2PE by a factor 6 not entirely convincing, as after dividing by the 6.7 larger volume of rod OS A2PE concentration is almost identical in the OS of rod-dominant and cone-dominant retinas.
In addition, the relevance of rod saturation for at-RAL generation has been questioned by recent data showing that rod saturation indicates their cGMP-sensitive channels are closed, not that all rhodopsins have been bleached. Indeed, rods adapted to continuous bright light for over 20 minutes may respond to bright light stimuli, indicating that cones and rods do generate at-RAL in bright light.
In the revised version, we discussed these points and quoted the relevant papers (l. 372-435).
Q12: Page 8 line 356-358. This does not prove that A2E accumulation is not driving pathology, this evidence in [69] shows mechanistically how A2E can cause photoreceptor loss, and how this may be alleviated by vitamin E, protecting against A2E-oxidation.
A12: We agree with the reviewer that preventing A2E photodegradation by vitamin E does not indicate that A2E is not driving the pathology. Again, the point is not whether but how. In the revised version, we discussed the evidence gathered from quantitative fundus autofluorescence on lipofuscin distribution across the macula and nearby regions and the protection afforded by vitamin E in mice KO for Abca4. (l. 436-480)
Q13: Page 8 line 376. This needs to be expanded to counter current work suggesting dysregulation of the retinoid cycle drives pathology in the RPE by Radu’s group. doi: 10.1073/pnas.1802519115. There is also the consideration that without a macula, as seen in abca4 -/- mouse models lipofuscin is detected but not thinning of the retina as the photoreceptor and RPE does not atrophy.
A13: We agree with the reviewer that the paragraphs preceding the short statement in l. 376 of the original version was insufficient, and the point had to be expanded.
In the revised version, we discussed the role of blue light impinging on the macula to assess whether it may represent the critical factor linking A2E photodegradation to macular damage. The data of Radu et al., 2016 indicate vitamin E suppresses bisretinoids photodegradation, the formation of short-chain adducts derived from bisretinoids photodegradation, and reduced retinal thinning. These data indicate a link between bisretinoids photodegradation and retinal damage in animal models lacking foveal cones. To evaluate the possible impact of vitamin E protection on macular cones, we discussed the evidence addressing whether vitamin E may prevent AMD onset, i.e., the triggering of macular damage in early AMD, or it may afford protection from the progression from intermediate to advanced AMD. Several randomized and controlled trials did not find evidence for vitamin E protecting from early AMD. Still, the AREDS2 study found efficacy in preventing the progression from intermediate to advanced AMD when vitamin E was administered at very high doses along with other vitamins and minerals antioxidants (l. 490- 535).
Q14: Page 9 line 416. This section uses “11-cis RAL” where previously uses “11 cis-RAL”.
A14: In the revised version, 11-cis RAL has been corrected in the heading and text of section 4.1 (l.695-769).
Q15: Page 10. Figure 3. Panel d is identified as c.
A15: In the revised version, the labeling in Figure has been corrected from c to d.
Q16: Page 11 line 445-447 11-cis RAL and 11 cis-RAL are both being used.
A16: In the revised version, 11-cis RAL has been corrected to 11cis-RAL and 11cis-ROL (l.694-769).
Q17: Page 11 line 451 “1-cis RAL”
A17: In the revised version, the text has been amended to read, “….involved in the 11cis-RAL synthesis …” (l. 730).
Q18: Page 17 line 724 “that translational control on RPE65” should be “that translational control of RPE65”.
A18: In the revised version, the sentence reads “……considering that translation control of RPE65 may represent ………….” (l. 1012).
Q19: Page 17 line 761-762 “11-cis RAL” and “11-cis-ROL”. Should these be “at-RAL” and “at-ROL”.
A19: In the revised version, the sentence has been changed to “Although defects in the conversion of at-RAL released from bleached opsin to 11cis-ROL have been associated…….” (l.1049-1050)
Q20: Page 17 line 761-764 (some further considerations mentioned as per page 8) Whilst RPE atrophy does proceed photoreceptor loss in stargardt’s disease there is a clear implication of retinoid dysregulation driving the disease since it is caused by reduced function of ABCA4. Current work does show ABCA4 may have additional roles, including in the RPE, but these still appear to be related to excessive all-trans-retinal or NRPE. There would need to be a proposed link in the RPE dysregulation / reduced metabolism caused which makes sense in the context of this genetic background, which has previously provided support to the bisretinoid driving the RPE loss and subsequent photoreceptor loss. There may be a higher metabolic demand on the macula where any dysfunction in the RPE compromises the ability to maintain itself and the photoreceptors it supports.
A20: We acknowledge that retinoid dysregulation may impact endosome trafficking and lysosome operation in the RPE. The revised version now includes a new section (3.2 Dysregulated lipid metabolism in RPE and photoreceptors’ demise) discussing the role of dysregulated endosome trafficking and lysosome dysfunction in macular dystrophies, including Stargardt 4 disease due to PROM1 defects linked to endosome processing and unrelated to bisretinoids dysregulation. In section 3.1, the impact of ABCA4 in RPE cells is presented and discussed, along with the impact of lysosome alkalinization and complement activation. Section 3.1 also discusses the role of RPE peroxisomes in lysosome function and the impact of DHA in retinal dystrophies, a topic relevant to macular dystrophies considering the difference between rods and cones in their DHA content. In the last part of section 3.1, we have expanded the point to address the higher energetic requirement of foveal cones. (l. 554-672).
Reviewer 4 Report
Comments and Suggestions for Authors
This review aims to provide an overview of the visual cycles in rod and cone cells across vertebrate and invertebrate species, along with their regulation during eye development. The authors gathered information from 144 relevant papers to offer a comprehensive perspective on this subject; however, the content and perspective presented in this work appear premature, and the language requires improvement
Minors:
1. The authors have introduced new abbreviations for terms that already have established common names in the field. For instance, they use "IPRB" for "interphotoreceptor binding proteins" instead of the widely recognized abbreviation, "IRBP" (page 5).
2. It should be noted that A2E is primarily synthesized in the retinal pigment epithelium (RPE) rather than the retina, as indicated in lines 330-335.
3. It is recommended to include a diagram illustrating the developmental events of retinal cells in section 4 for enhanced clarity and understanding.
Comments on the Quality of English Language
This review aims to provide an overview of the visual cycles in rod and cone cells across vertebrate and invertebrate species, along with their regulation during eye development. The authors gathered information from 144 relevant papers to offer a comprehensive perspective on this subject; however, the content and perspective presented in this work appear premature, and the language requires improvement
Minors:
1. The authors have introduced new abbreviations for terms that already have established common names in the field. For instance, they use "IPRB" for "interphotoreceptor binding proteins" instead of the widely recognized abbreviation, "IRBP" (page 5).
2. It should be noted that A2E is primarily synthesized in the retinal pigment epithelium (RPE) rather than the retina, as indicated in lines 330-335.
3. It is recommended to include a diagram illustrating the developmental events of retinal cells in section 4 for enhanced clarity and understanding.
Author Response
This review aims to provide an overview of the visual cycles in rod and cone cells across vertebrate and invertebrate species, along with their regulation during eye development. The authors gathered information from 144 relevant papers to offer a comprehensive perspective on this subject; however, the content and perspective presented in this work appear premature, and the language requires improvement
We thank the reviewer for the criticisms and constructive suggestions. Section 3 has been expanded to strengthen our points in the revised version, and several typos and errors have been corrected.
Minors:
Q1: The authors have introduced new abbreviations for terms that already have established common names in the field. For instance, they use "IPRB" for "interphotoreceptor binding proteins" instead of the widely recognized abbreviation, "IRBP" (page 5).
A1: In the revised version, all abbreviations are consistent, including the standard term IRBP.
Q2: It should be noted that A2E is primarily synthesized in the retinal pigment epithelium (RPE) rather than the retina, as indicated in lines 330-335.
A2: In the revised version, the sentence now reads, “A2E results from the reaction of a second at-RAL with NRPE to generate A2PE [75,76], which is hydrolyzed to A2E in the RPE [75,76]. Upon A2E exposure to light, the mixture of A2E isomers in the RPE [76] indicates that retina-generated A2PE may be taken up by RPE as a result of disk-shedding and undergo further conversion in the RPE in response to light.” (l.356-361).
Q3. It is recommended to include a diagram illustrating the developmental events of retinal cells in section 4 for enhanced clarity and understanding.
A3: In the revised version, figure 4 includes a diagram in panel d with the timeline of the main events occurring during retinal development between FW 4 and FW12.
Reviewer 5 Report
Comments and Suggestions for Authors
The manuscript written by Massimiliano Andreazzoli et al. talks about the role of retinal cells in retinoid synthesis from development to the adult stage. Although the manuscript was written well with a solid background, the content is not properly organized. Additionally, adding the latest information will add better visibility to the current version. Major concerns are.
- The section “Advantages and disadvantages of the retinoid cycle” should come after introducing the retinoid cycle.
- Instead of discussion, it should be a conclusion, as the authors are not presenting any data here.
- Line 469: Author is talking about the overexpression of RPE65 and saying it is non-significant. However, the FDA recently approved RPE65 gene therapy for inherited retinal dystrophy. This should be included, as it’s a milestone for gene therapy in the eye.
- Line 62-63, Please introduce the full form of the word RPE as it’s introduced. FW is not expanded anywhere in the article. Similarly, please check other words too.
- All sections are disconnected, for example, sections 4.1 and 4.3 are similar and should be merged as the authors are talking about retinol synthesis in RPE cells. I believe the article can be organized in a better way to make it interesting for the audience. For example: Introduction, Eye structure and various cells, visual cycle: retinoids and its regulation, Retinoids in diseased conditions and their effect on viability, Therapeutics intervention to retinoid cycle.
- The author needs to decide for the title of the manuscript, as I feel the current title is not appropriate. I would suggest this: Retinoid synthesis regulation by retinal cells in health and disease conditions. For retinoids in development, authors should refer to PMID 10893430.
Moderate editing of the English language is required.
Author Response
The manuscript written by Massimiliano Andreazzoli et al. talks about the role of retinal cells in retinoid synthesis from development to the adult stage. Although the manuscript was written well with a solid background, the content is not properly organized. Additionally, adding the latest information will add better visibility to the current version. Major concerns are.
We want to thank the reviewer for the constructive comments. In the revised version, we partially reorganized the text. We expanded section 3 to include the latest developments relevant to the understanding of the mechanisms involved in macular dystrophies. Below is a point-by-point reply to specific points raised by the reviewer:
Q1: The section “Advantages and disadvantages of the retinoid cycle” should come after introducing the retinoid cycle.
A1: In the revised version, the section 2 heading now reads “The retinoids cycle” and its content focuses on the retinoid cycle. (l. 74). The section 3 headings now reads “Retinoids' cycle and photoreceptors' viability”, and its content focuses on the impact of the retinoid cycle on photoreceptor viability. (l. 309)
Q2: Instead of discussion, it should be a conclusion, as the authors are not presenting any data here.
A2: In the revised version, the section 5 heading is now “Conclusion” (l. 1041)
Q3: Line 469: Author is talking about the overexpression of RPE65 and saying it is non-significant. However, the FDA recently approved RPE65 gene therapy for inherited retinal dystrophy. This should be included, as it’s a milestone for gene therapy in the eye.
A3: In the revised version, we added a sentence referring to using an adenovirus for the gene therapy of biallelic pathogenic mutation of RPE65, “As noted by [178], this control may represent a safety mechanism for patients treated with viral vectors carrying RPE65 for the first approved gene therapy of biallelic pathogenic variants of RPE65 [186-188]. It is unclear whether the translational control of RPE65 reported for lentivirus-transfected RPE65 [178] will also operate for the adenovirus-transfected RPE65 used for gene therapy [186-188].” (l. 766-769)
Q4: Line 62-63, Please introduce the full form of the word RPE as it’s introduced. FW is not expanded anywhere in the article. Similarly, please check other words too.
A4: In the revised version, the retinal pigment epithelium (RPE) is defined in l. 64-65, and the term fetal week (FW) is defined in l. 772-773.
Q5: All sections are disconnected, for example, sections 4.1 and 4.3 are similar and should be merged as the authors are talking about retinol synthesis in RPE cells. I believe the article can be organized in a better way to make it interesting for the audience. For example: Introduction, Eye structure and various cells, visual cycle: retinoids and its regulation, Retinoids in diseased conditions and their effect on viability, Therapeutics intervention to retinoid cycle.
A5: Section 4.1 focused on the mechanism regulating the synthesis of vitamin A derivatives involved in the retinoid cycle, such as 11cis-RAL and at-RAL. Section 4.3 focused on vitamin A derivatives involved in photoreceptor development, such as 9cis-RAL, 9cis-RA, and at-RA. We believe that the retinal development in section 4.2 must precede section 4.3, and merging sections 41.1 and 4.3 wouldn’t help the reader. Regarding the reorganization of the manuscript, we didn’t cover the therapeutic interventions in the retinoid cycle. Moreover, placing the content of section 4.2 (eye structure) before section 2 (the retinoid cycle in the revised version) may not help the reader with the retinoid chemistry in section 2. It would be too far from the cell biology content of section 4.3.
Q6: The author needs to decide for the title of the manuscript, as I feel the current title is not appropriate. I would suggest this: Retinoid synthesis regulation by retinal cells in health and disease conditions. For retinoids in development, authors should refer to PMID 10893430.
A6: We met the reviewer's suggestion by changing the revised version's title to “Retinoid synthesis regulation by retinal cells in health and disease”.
In introducing section 4.3, we added the reference suggested by the reviewer. (l.963-964)
Round 2
Reviewer 3 Report
Comments and Suggestions for Authors
Comments and Suggestions for Authors
The perspective discusses the regulation of 11-cis-retinal concerning photoreceptor development and survival. This shift focuses from the current perspective that bis-retinoid products generated from excess all-trans-retinal following visual transduction drive the macula degeneration observed in retinal diseases such as Stargardts. The changes suggested to include further considerations in the discussion have been addressed comprehensively.
All identified points have been addressed. Only some new minor typos in the re-draft for attention below. Otherwise, much improvement has been made particularly in section 3.
Page 5 line 193 After [35] there is (REF).
Page 10 line 450 “Stargard” change to “Stargardt”
Page 21 line 947 “effectively augment the surface area of the RPE” is different font to rest of text.
Author Response
We want to thank the reviewer for the careful revision. In the following the reply to the specific points:
Q1: Page 5 line 193 After [35] there is (REF).
A1:In the revised version, (REF) has been removed (p. 4, l. 177)
Q2: Page 10 line 450 “Stargard” change to “Stargardt”
A2: In the revised version, the typo has been corrected (p. 11, l. 540)
Q3: Page 21 line 947 “effectively augment the surface area of the RPE” is different font to rest of text.
A3: In the revised version, the character type has been corrected (p. 20, l. 971)
Reviewer 4 Report
Comments and Suggestions for Authors
This revision has addressed most of my concerns, and I see a great improvement in this manuscript. More references were added, too. After minor revisions, this work is now ready for publication.
1. Since the title is changed to focus on the regulation of retinoid synthesis in health and disease, the contents about the complex eye, c—and r—photoreceptors (UV) seem inappropriate for this manuscript.
2. Please correct the IPRBP to IRBP in Figure 2.
Author Response
We want to thank the reviewer for the constructive comments and for the careful revision of the text. The reply to specific points is in the following:
Q1: Since the title is changed to focus on the regulation of retinoid synthesis in health and disease, the contents about the complex eye, c—and r—photoreceptors (UV) seem inappropriate for this manuscript.
A1: We met reviewer #5’s suggestion to change the title, which we thought was consistent with the expansion of section 3 at the request of reviewer #3. In our opinion, the title change does not make the section on rhabdomeric and ciliary receptors redundant. The section in the introduction is recalled throughout the paper when we discuss the relevance of pigment photoconversion in vertebrates. Removing these lines from the introduction may not significantly shorten the text. Still, it could be difficult for the reader to understand the notion and the relevance of photoconversion we discuss in the following sections.
Q2: Please correct the IPRBP to IRBP in Figure 2.
A2: In the revised version, the typo has been corrected (note that in the revised version, Figure 2 is now Figure 3, p.5, l. 182-231)
Reviewer 5 Report
Comments and Suggestions for Authors
The authors have responded to my comments and improved the manuscript. I still have corrections.
Heading 3 and its subheading 3.1 title are almost the same. 3.1 should now be named "Studies in animal models with a defective retinoid cycle.".
Additionally, now a table form is required for all genetically deleted animal models mentioned in the entire manuscript, along with their phenotypes.
Section 4.1 heading should be rewritten as "The transcriptional regulation of the retinoid cycle. Section 4.2 should come earlier, as it talks about the development of eye structure. This must be a separate section with the heading; Development of Eye and Photoreceptors.
I have also mentioned in my first comments that the section containing the development of the eye, cells, and photoreceptors should be moved after the introduction.
Comments on the Quality of English LanguageMinor editing is required.
Author Response
We want to thank the reviewer for the constructive comments and helpful suggestions. In the two revisions, we have met most of the requests. Even when we disagree on some suggestions, we are aware of the merit of the reviewer's points and have been striving to partially meet the suggestions. In the following we provide the point-by-point reply to the specific points:
Q1: Heading 3 and its subheading 3.1 title are almost the same. 3.1 should now be named "Studies in animal models with a defective retinoid cycle."
A1: We appreciate the merit of the reviewer’s suggestion. However, the suggested change may not fully catch the content of a section reporting studies in reconstituted membranes, transgenic animals, and patients. To meet the reviewer’s suggestion to differentiate the subheading, we have modified the subheading, clarifying that the section will focus on the mechanisms linking retinoid cycle defects to reduced photoreceptor viability. We propose as section 3.1 title of the revised version: “Mechanisms linking retinoid cycle defects to impaired photoreceptors’ viability.” (p. 9, l. 409)
Q2: Additionally, now a table form is required for all genetically deleted animal models mentioned in the entire manuscript, along with their phenotypes.
A2: In the revised version, Table 2 in section 3.1 now reports the transgenic mouse models, the function of deleted genes, and the mouse phenotype (p. 10, l. 453-484).
Q3: Section 4.1 heading should be rewritten as "The transcriptional regulation of the retinoid cycle. Section 4.2 should come earlier, as it talks about the development of eye structure. This must be a separate section with the heading; Development of Eye and Photoreceptors.
A3: In the revised version, the section on eye and retinal development is now 4.1 and is entitled “Development of retinal cells involved in 11cis-RAL synthesis and isomerization” (revised version p.16, l. 787).
Section 4.1 now precedes section 4.2, entitled “Regulation of the retinoid cycle.” We did not take the suggestion to include the term transcriptional control, as we consider it too restricted. Indeed, in section 4.2 of the revised version, we discuss the evidence of transcriptional, translational, and posttranslational control over RPE65 activity. According to the title of section 4.2, we have maintained section 4.3, dealing with the possible role of 9cis-RAL in regulating opsin transcription, separate from 4.2, dealing with 11cis-RAL synthesis regulation.
Q4: I have also mentioned in my first comments that the section containing the development of the eye, cells, and photoreceptors should be moved after the introduction.
A4: In the first revision, the reviewer suggested a different organization of the perspective. However, although we did not mention it in our previous reply, the suggestion did not state where to place the section on retinal cell development. In a separate comment, the suggestion for retinal development was to quote a review. Accordingly, we introduced section 4.3 by quoting the review suggested.
Nevertheless, considering that the review is not specific to the eye, and more evidence has been accumulating on retinal development since July 2000, we still believe that the regulation of retinoid synthesis during retinal development in section 4 is a timely contribution, worth a section in this perspective.
We politely disagree with the suggestion to split the section on retinal development from the regulation of retinoid synthesis. In our opinion, the suggested change would break the logical flow of the perspective. Sections 2 and 3 present evidence that RPE and MGCs generate 11cis-RAL and that either too low or too high 11cis-RAL generation negatively affects photoreceptors and RPE cell viability. Section 4 unfolds this concept during development by presenting evidence of the match between the functional maturation of RPE and photoreceptor cells despite their differing birthdates. In other words, section 4 is not just a section on retinal anatomy that would be better placed after the introduction. In our view, the combined analysis of data on RPE and photoreceptor development and their functional maturation in sections 4.1 and 4.2 is a timely and original contribution. To date, no review or paper has covered yet the coordinated maturation of 11cis-RAL synthesis in RPE cells with the amount of opsin binding sites in photoreceptors. The relevance of this point is consistent with the 2023 PNAS paper by Xue et al., which we have quoted and discussed [58], showing that reducing 11cis-RAL synthesis after rod cells have degenerated improves cone survival.
Although we disagree with the reviewer's suggestion to split section 4 in the revised version, to meet her/his suggestion and enhance text comprehension, we introduced a new figure (Figure 2 text, p.3, l 123-128—Figure 2 in page 4, l. 146-160 of the revised version) showing the main cell types involved in the retinoid cycle.
Round 3
Reviewer 5 Report
Comments and Suggestions for Authors
The authors have responded to my comments. I just wanted to improve the readability of the manuscript. In the first version, the title was on development to adulthood; based on that, I might have given that suggestion. As the authors have changed the title, everything is okay now.
I am not happy with the format of the table legend. This is not the standard way of writing the legends on a table. Please look at other published articles and rewrite them.
Comments on the Quality of English Language
Minor editing is required.
Author Response
We appreciate the reviewer's comments on this and the previous revisions and feel they have improved the paper's readability. Accordingly, we have modified the Table 2 legend to meet the Cells standard (l.483-484 of the revised version).